# DiT-3D: Exploring Plain Diffusion Transformers for 3D Shape Generation

**Shentong Mo**[1] **Enze Xie**[2*] **Ruihang Chu**[3] **Lewei Yao**[2]
**Lanqing Hong**[2] **Matthias Nießner**[4] **Zhenguo Li**[2]

[1]MBZUAI, [2]Huawei Noah's Ark Lab, [3]CUHK, [4]TUM
https://DiT-3D.github.io

## Abstract

Recent Diffusion Transformers (*e.g.* DiT [1]) have demonstrated their powerful effectiveness in generating high-quality 2D images. However, it is still being determined whether the Transformer architecture performs equally well in 3D shape generation, as previous 3D diffusion methods mostly adopted the U-Net architecture. To bridge this gap, we propose a novel Diffusion Transformer for 3D shape generation, namely DiT-3D, which can directly operate the denoising process on voxelized point clouds using plain Transformers. Compared to existing U-Net approaches, our DiT-3D is more scalable in model size and produces much higher quality generations. Specifically, the DiT-3D adopts the design philosophy of DiT [1] but modifies it by incorporating 3D positional and patch embeddings to adaptively aggregate input from voxelized point clouds. To reduce the computational cost of self-attention in 3D shape generation, we incorporate 3D window attention into Transformer blocks, as the increased 3D token length resulting from the additional dimension of voxels can lead to high computation. Finally, linear and devoxelization layers are used to predict the denoised point clouds. In addition, our transformer architecture supports efficient fine-tuning from 2D to 3D, where the pre-trained DiT-2D checkpoint on ImageNet can significantly improve DiT-3D on ShapeNet. Experimental results on the ShapeNet dataset demonstrate that the proposed DiT-3D achieves state-of-the-art performance in high-fidelity and diverse 3D point cloud generation. In particular, our DiT-3D decreases the 1-Nearest Neighbor Accuracy of the state-of-the-art method by 4.59 and increases the Coverage metric by 3.51 when evaluated on Chamfer Distance.

## 1 Introduction

In recent times, there has been a growing interest in exploring the potential of diffusion transformers for high-fidelity image generation, as evinced by a series of scholarly works [1, 2, 3, 4]. Notably, a seminal work by Peebles et al. [1] proposed the replacement of the widely-used U-Net backbone with a scalable transformer. Specifically, the proposed method operates on latent patches by training latent 2D diffusion models. However, the efficacy of plain diffusion transformers for 3D shape generation has yet to be explored, as most existing 3D diffusion approaches continue to adopt the U-Net backbone.

Generating high-fidelity point clouds for 3D shape generation is a challenging and significant problem. Early generative methods [5, 6, 7] addressed this problem by directly optimizing heuristic loss objectives, such as Chamfer Distance (CD) and Earth Mover's Distance (EMD). More recent works [8, 9, 10, 11] have explored the usage of the generative adversarial network (GAN)-based

---

*Corresponding author.

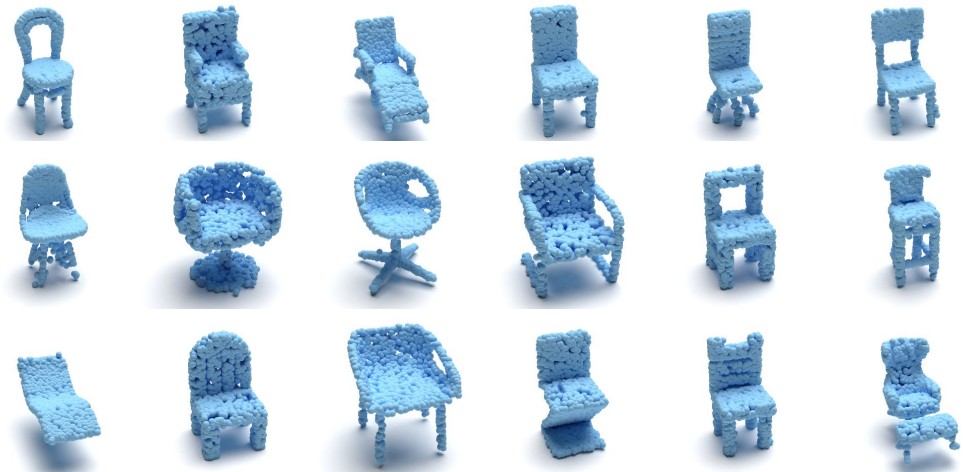

Figure 1: Examples of high-fidelity and diverse 3D point clouds produced from DiT-3D.

and flow-based models to generate 3D point clouds from a probabilistic perspective. Recently, researchers [12, 13, 14, 15] have turned to various denoising diffusion probabilistic models (DDPMs) to generate entire shapes from random noise. For instance, PVD [12] employed the point-voxel representation of 3D shapes as input to DDPMs. They reversed the diffusion process from observed point clouds to Gaussian noise by optimizing a variational lower bound to the likelihood function. Recently, the Diffusion Transformer (DiT) [1, 2] has been shown to surpass the U-Net architecture in 2D image generation, owing to its simple design and superior generative performance. Consequently, we investigate the potential of the Diffusion Transformer for 3D generation. However, extending the 2D DiT to 3D poses two significant challenges: (1) Point clouds are intrinsically unordered, unlike images where pixels are ordered; and (2) The tokens in 3D space have an additional dimension compared to 2D images, resulting in a substantial increase in computational cost.

This work introduces DiT-3D, a novel diffusion transformer architecture designed for 3D shape generation that leverages the denoising process of DDPM on 3D point clouds. The proposed model inherits the simple design of the modules in DiT-2D, with only minor adaptations to enable it to generalize to 3D generation tasks. To tackle the challenge posed by the unordered data structure of point clouds, we convert the point cloud into a voxel representation. DiT-3D employs 3D positional embedding and 3D patch embedding on the voxelized point clouds to extract point-voxel features and effectively process the unordered data. Furthermore, to address the computational cost associated with a large number of tokens in 3D space, we introduce a 3D window attention operator instead of the vanilla global attention in DiT-2D. This operator significantly reduces training time and memory usage, making DiT-3D feasible for large-scale 3D generation tasks. Finally, we utilize linear and devoxelization layers to predict the noised point clouds in the reversed process to generate final 3D shapes.

In order to address the computational cost associated with a large number of tokens in 3D space, we also introduce a parameter-efficient tuning method to utilize the pre-trained DiT-2D model on ImageNet as initialization for DiT-3D (window attention shares the same parameters with vanilla attention). Benefiting from the substantial similarity between the network structure and parameters of DiT-3D and DiT-2D, the representations learned on ImageNet significantly improve 3D generation, despite the significant domain disparity between 2D images and 3D point clouds. To our knowledge, we are the first to achieve parameter-efficient fine-tuning from 2D ImageNet pre-trained weights for high-fidelity and diverse 3D shape generation. In particular, we highly decrease the training parameters from 32.8MB to only 0.09MB.

We present a comprehensive evaluation of DiT-3D on a diverse set of object classes in the ShapeNet benchmark, where it achieves state-of-the-art performance compared to previous non-DDPM and DDPM-based 3D shape generation methods. Qualitative visualizations further emphasize the efficacy of DiT-3D in generating high-fidelity 3D shapes. Extensive ablation studies confirm the significance of 3D positional embeddings, window attention, and 2D pre-training in 3D shape generation. Moreover, we demonstrate that DiT-3D is easily scalable regarding patch sizes, voxel sizes, and model sizes. Our

findings align with those of DiT-2D, where increasing the model size leads to continuous performance improvements. In addition, our parameter-efficient fine-tuning from DiT-2D ImageNet pre-trained weights highly decreases the training parameters while achieving competitive generation performance. By only training 0.09MB parameters of models from the source class to the target class, we also achieve comparable results of quality and diversity in terms of all metrics.

Our main contributions can be summarized as follows:

- We present DiT-3D, the first plain diffusion transformer architecture for point cloud shape generation that can effectively perform denoising operations on voxelized point clouds.

- We make several simple yet effective modifications on DiT-3D, including 3D positional and patch embeddings, 3D window attention, and 2D pre-training on ImageNet. These modifications significantly improve the performance of DiT-3D while maintaining efficiency.

- Extensive experiments on the ShapeNet dataset demonstrate the state-of-the-art superiority of DiT-3D over previous non-DDPM and DDPM baselines in generating high-fidelity shapes.

## 2 Related Work

**3D Shape Generation.** 3D shape generation aims to synthesize high-fidelity point clouds or meshes using generative models, such as variational autoencoders [16, 17, 18], generative adversarial networks [19, 8, 20], and normalized flows [9, 10, 11]. Typically, PointFlow [9] utilized a probabilistic framework based on the continuous normalizing flow to generate 3D point clouds from two-level hierarchical distributions. ShapeGF [21] trained a score-matching energy-based network to learn the distribution of points across gradient fields using Langevin dynamics. More recently, GET3D [14] leveraged a signed distance field (SDF) and a texture field as two latent codes to learn a generative model that directly generates 3D meshes. In this work, we mainly focus on denoising diffusion probabilistic models for generating high-fidelity 3D point clouds starting from random noise, where point and shape distributions are not separated.

**Diffusion Models.** Diffusion models [22, 23, 24] have been demonstrated to be effective in many generative tasks, such as image generation [25], image restoration [26], speech generation [27], and video generation [28]. Denoising diffusion probabilistic models (DDPMs) [22, 23] utilized a forward noising process that gradually adds Gaussian noise to images and trained a reverse process that inverts the forward process. In recent years, researchers [29, 12, 13, 30, 15, 31, 32] have tried to explore diverse pipelines based on diffusion probabilistic models to achieve 3D shape generation. For example, PVD [12] applied DDPM based on PVCNNs [33] on the point-voxel representation of 3D shapes with structured locality into point clouds. To improve the generation quality, LION [13] used two DDPMs to learn a hierarchical latent space based on a global shape latent representation and a point-structured latent space separately. Different from them, we will solve the 3D shape generation problem in our approach by designing a plain transformer-based architecture backbone to replace the U-Net backbone for reversing the diffusion process from observed point clouds to Gaussian noise. Meanwhile, our 3D plain diffusion transformer supports multi-class training with learnable class embeddings as the condition and parameter-efficient fine-tuning with modality and domain transferability differ from DDPM-based 3D generation approaches discussed above.

**Transformers in Diffusion Generation.** Diffusion Transformers [1, 2, 3, 4] have recently shown their impressive capacity to generate high-fidelity images. For instance, Diffusion Transformer (DiT) [1] proposed a plain diffusion Transformer architecture to learn the denoising diffusion process on latent patches from a pre-trained pre-trained variational autoencoder model in Stable Diffusion [34]. U-ViT [2] incorporated all the time, condition, and noisy image patches as tokens and utilized a Vision transformer(ViT) [35]-based architecture with long skip connections between shallow and deep layers. More recently, UniDiffuser [3] designed a unified transformer for diffusion models to handle input types of different modalities by learning all distributions simultaneously. While those diffusion transformer approaches achieve promising performance in 2D image generation, how a plain diffusion transformer performs on 3D shape generation is still being determined. In contrast, we develop a novel plain diffusion transformer for 3D shape generation that can effectively perform denoising operations on voxelized point clouds. Furthermore, the proposed DiT-3D can support parameter-efficient fine-tuning with transferability across modality and domain.

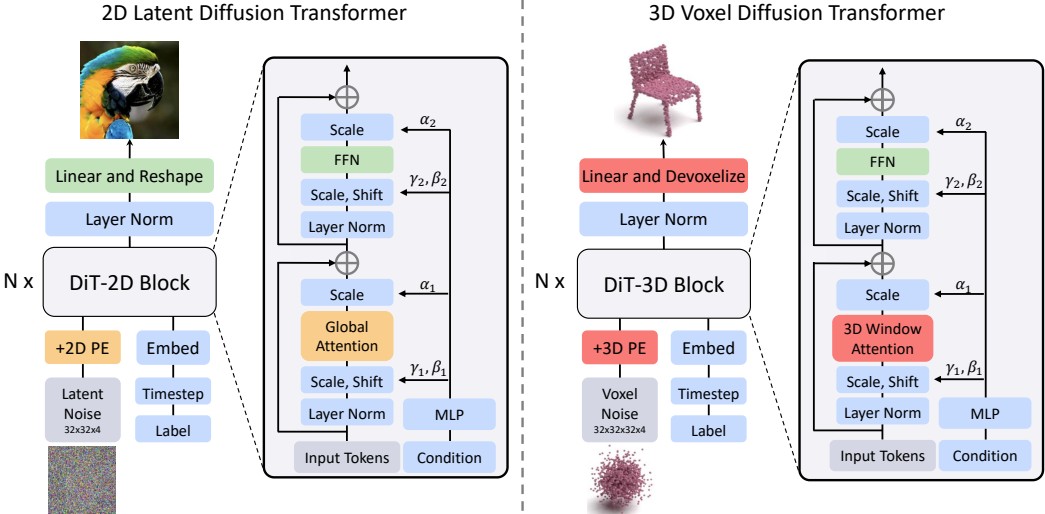

Figure 2: Illustration of the proposed Diffusion Transformers (DiT-3D) for 3D shape generation. The plain diffusion transformer takes voxelized point clouds as input, and a patchification operator is used to generate token-level patch embeddings, where 3D positional embeddings are added together. Then, multiple transformer blocks based on 3D window attention extract point-voxel representations from all input tokens. Finally, the unpatchified voxel tensor output from a linear layer is devoxelized to predict the noise in the point cloud space.

## 3 Method

Given a set of 3D point clouds, we aim to learn a plain diffusion transformer for synthesizing new high-fidelity point clouds. We propose a novel diffusion transformer that operates the denoising process of DDPM on voxelized point clouds, namely DiT-3D, which consists of two main modules: Design DiT for 3D Point Cloud Generation in Section 3.2 and Efficient Modality/Domain Transfer with Parameter-efficient Fine-tuning in Section 3.3.

### 3.1 Preliminaries

In this section, we first describe the problem setup and notations and then revisit denoising diffusion probabilistic models (DDPMs) for 3D shape generation and diffusion transformers on 2D images.

**Problem Setup and Notations.** Given a set $\mathcal{S} = \{\mathbf{p}_i\}_{i=1}^{S}$ of 3D shapes with $M$ classes, our goal is to train a plain diffusion transformer from these point clouds for generating high-fidelity point clouds. For each point cloud $\mathbf{p}_i$, we have $N$ points for $x, y, z$ coordinates, that is $\mathbf{p}_i \in \mathbb{R}^{N \times 3}$. Note that we have a class label for the 3D shape $\mathbf{p}_i$, which is denoted as $\{y_i\}_{i=1}^{M}$ with $y_i$ for the ground-truth category entry $i$ as 1. During the training, we take the class label as input to achieve classifier-free guidance in conditional diffusion models, following the prior diffusion transformer (*i.e.*, DiT [1]) on images.

**Revisit DDPMs on 3D Shape Generation.** To solve the 3D shape generation problem, previous work [12] based on denoising diffusion probabilistic models (DDPMs) define a forward noising process that gradually applies noise to real data $\mathbf{x}_0$ as $q(\mathbf{x}_t|\mathbf{x}_{t-1}) = \mathcal{N}(\mathbf{x}_t; \sqrt{1 - \beta_t}\mathbf{x}_{t-1}, \beta_t\mathbf{I})$, where $\beta_t$ is a Gaussian noise value between 0 and 1. In particular, the denoising process produces a series of shape variables with decreasing levels of noise, denoted as $\mathbf{x}_T, \mathbf{x}_{T-1}, ..., \mathbf{x}_0$, where $\mathbf{x}_T$ is sampled from a Gaussian prior and $\mathbf{x}_0$ is the final output. With the reparameterization trick, we can have $\mathbf{x}_t = \sqrt{\bar{\alpha}_t}\mathbf{x}_0 + \sqrt{1 - \bar{\alpha}_t}\boldsymbol{\epsilon}$, where $\boldsymbol{\epsilon} \sim \mathcal{N}(\mathbf{0}, \mathbf{I})$, $\alpha_t = 1 - \beta_t$, and $\bar{\alpha}_t = \prod_{i=1}^{t} \alpha_i$.

For the reverse process, diffusion models are trained to learn a denoising network $\boldsymbol{\theta}$ for inverting forward process corruption as $p_{\boldsymbol{\theta}}(\mathbf{x}_{t-1}|\mathbf{x}_t) = \mathcal{N}(\mathbf{x}_{t-1}; \boldsymbol{\mu}_{\boldsymbol{\theta}}(\mathbf{x}_t, t), \sigma_t^2\mathbf{I})$. The training objective is to maximize a variational lower bound of the negative log data likelihood that involves all of $\mathbf{x}_0, ..., \mathbf{x}_T$

as

$$\mathcal{L} = \sum_t -p_{\boldsymbol{\theta}}(\mathbf{x}_0|\mathbf{x}_1) + \mathcal{D}_{\text{KL}}(q(\mathbf{x}_{t-1}|\mathbf{x}_t, \mathbf{x}_0)||p_{\boldsymbol{\theta}}(\mathbf{x}_{t-1}|\mathbf{x}_t))) \tag{1}$$

where $\mathcal{D}_{\text{KL}}(\cdot||\cdot)$ denotes the KL divergence measuring the distance between two distributions. Since both $p_{\boldsymbol{\theta}}(\mathbf{x}_{t-1}|\mathbf{x}_t))$ and $q(\mathbf{x}_{t-1}|\mathbf{x}_t, \mathbf{x}_0)$ are Gaussians, we can reparameterize $\boldsymbol{\mu}_{\boldsymbol{\theta}}(\mathbf{x}_t, t)$ to predict the noise $\boldsymbol{\epsilon}_{\boldsymbol{\theta}}(\mathbf{x}_t, t)$. In the end, the training objective can be reduced to a simple mean-squared loss between the model output $\boldsymbol{\epsilon}_{\boldsymbol{\theta}}(\mathbf{x}_t, t)$ and the ground truth Gaussian noise $\boldsymbol{\epsilon}$ as: $\mathcal{L}_{\text{simple}} = \|\boldsymbol{\epsilon} - \boldsymbol{\epsilon}_{\boldsymbol{\theta}}(\mathbf{x}_t, t)\|^2$. After $p_{\boldsymbol{\theta}}(\mathbf{x}_{t-1}|\mathbf{x}_t))$ is trained, new point clouds can be generated by progressively sampling $\mathbf{x}_{t-1} \sim p_{\boldsymbol{\theta}}(\mathbf{x}_{t-1}|\mathbf{x}_t))$ by using the reparameterization trick with initialization of $\mathbf{x}_T \sim \mathcal{N}(\mathbf{0}, \mathbf{I})$.

**Revisit Diffusion Transformer (DiT) on 2D Image Generation.** To generate high-fidelity 2D images, DiT proposed to train latent diffusion models (LDMs) with Transformers as the backbone, consisting of two training models. They first extract the latent code $\mathbf{z}$ from an image sample $\mathbf{x}$ using an autoencoder with an encoder $f_{\text{enc}}(\cdot)$ and a decoder $f_{\text{dec}}(\cdot)$, that is, $\mathbf{z} = f_{\text{enc}}(\mathbf{x})$. The decoder is used to reconstruct the image sample $\hat{\mathbf{x}}$ from the latent code $\mathbf{z}$, *i.e.*, $\hat{\mathbf{x}} = f_{\text{dec}}(\mathbf{z})$. Based on latent codes $\mathbf{z}$, a latent diffusion transformer with multiple designed blocks is trained with time embedding $\mathbf{t}$ and class embedding $\mathbf{c}$, where a self-attention and a feed-forward module are involved in each block. Note that they apply patchification on latent code $\mathbf{z}$ to extract a sequence of patch embeddings and depatchification operators are used to predict the denoised latent code $\mathbf{z}$.

Although DDPMs achieved promising performance on 3D shape generation, they can only handle single-class training based on PVCNNs [33] as the encoder to extract 3D representations, and they cannot learn explicit class-conditional embeddings. Furthermore, we are not able to directly transfer their single-class pre-trained model to new classes with parameter-efficient fine-tuning. Meanwhile, we empirically observe that the direct extension of DiT [1] on point clouds does not work. To address this problem, we propose a novel plain diffusion transformer for 3D shape generation that can effectively achieve the denoising processes on voxelized point clouds, as illustrated in Figure 2.

## 3.2 Diffusion Transformer for 3D Point Cloud Generation

To enable denoising operations using a plain diffusion transformer, we propose several adaptations to 3D point cloud generation in Figure 2 within the framework of DiT [1]. Specifically, our DiT-3D model accepts voxelized point clouds as input and employs a patchification operator to generate token-level patch embeddings. We add 3D positional embeddings to these embeddings and extract point-voxel representations from all input tokens using multiple transformer blocks based on 3D window attention. Finally, we apply a devoxelized linear layer to the unpatchified voxel output, allowing us to predict the noise in the point cloud space.

**Denoising on Voxelized Point Clouds.** Point clouds are inherently unordered, unlike images where pixels follow a specific order. We encountered difficulty in our attempt to train a diffusion transformer on point coordinates due to the sparse distribution of points in the 3D embedding space. To address this issue, we decided to voxelize the point clouds into dense representations, allowing the diffusion transformers to extract point-voxel features. Our approach differs from DiT [1], which utilizes latent codes $\mathbf{z}$ to train the latent diffusion transformer. Instead, we directly train the denoising process on voxelized point clouds using the diffusion transformer. For each point cloud $\mathbf{p}_i \in \mathbb{R}^{N \times 3}$ with $N$ points for $x, y, z$ coordinates, we first voxelize it as input $\mathbf{v}_i \in \mathbb{R}^{V \times V \times V \times 3}$.

**3D Positional and Patch Embeddings.** With the voxel input $\mathbf{v}_i \in \mathbb{R}^{V \times V \times V \times 3}$, we introduce patchification operator with a patch size $p \times p \times p$ to generate a sequence of patch tokens $\mathbf{t} \in \mathbb{R}^{L \times 3}$. $L = (V/p)^3$ denotes the total number of patchified tokens. A 3D convolution layer is applied on patch tokens to extract patch embeddings $\mathbf{e} \in \mathbb{R}^{L \times D}$, where $D$ is the dimension of embeddings. To adapt to our voxelized point clouds, we add frequency-based sine-cosine 3D positional embeddings instead of the 2D version in DiT [1] to all input tokens. Based on these patch-level tokens, we introduce time embeddings $\mathbf{t}$ and class embeddings $\mathbf{c}$ as input to achieve multi-class training with learnable class embeddings as the condition, which differs from existing 3D generation approaches with U-Net as the backbone.

**3D Window Attention.** Due to the increased token length resulting from the additional dimension in 3D space, the computational cost of 3D Transformers can be significantly high. To address this issue, we introduce efficient 3D window attention into Transformer blocks blocks to propagate point-voxel

features in efficient memory usage. For the original multi-head self-attention process with each of the heads $Q, K, V$ have the same dimensions $L \times D$, where $L = (V/p)^3$ is the length of input tokens, we can have the attention operator as:

$$\text{Attention}(Q, K, V) = \text{Softmax}(\frac{QK^\top}{\sqrt{D_h}}V) \qquad (2)$$

where $D_h$ is the dimension size of each head. The computational complexity of this process is $\mathcal{O}(L^2)$, which will be largely expensive for high voxel resolutions. Inspired by [36], we extend the 2D window attention operator to a 3D one for 3D input tokens instead of vanilla global attention. This process uses a window size of $R$ to reduce the length of total input tokens as

$$\hat{K} = \text{Reshape}(\frac{L}{R^3}, D \cdot R^3)$$
$$K = \text{Linear}(D \cdot R^3, D)(\hat{K}) \qquad (3)$$

where $K$ is the input tokens to be reduced. Reshape $\left(\frac{L}{R^3}, D \cdot R^3\right)$ denotes to reshape $K$ to the one with shape of $\frac{L}{R^3} \times (D \cdot R^3)$, and $\text{Linear}(C_{in}, C_{out})(\cdot)$ denotes to a linear layer with a $C_{in}$-dimensional tensor as input and a $C_{out}$-dimensional tensor as output. Therefore, the new $K$ has the shape of $\frac{L}{R^3} \times D$. As a result, the complexity of the self-attention operator in Equation (2) is reduced from $\mathcal{O}(L^2)$ to $\mathcal{O}(\frac{L^2}{R^3})$. In our experiments, we set $R$ to $4$ in the default setting.

**Devoxelized Prediction.** Since the transformers blocks are implemented on voxelized point clouds, we can not directly use a standard linear decoder to predict the output noise $\epsilon_\theta(\mathbf{x}_t, t)$ from point clouds. In order to generate the output noise, we devoxelize output tokens from the linear decoder. We first apply the final layer norm and linearly decode each token into a $p \times p \times p \times L \times 3$ tensor, where $L$ is the total number of input tokens. Then we unpatchify the decoded token into a voxel tensor with the shape of $V \times V \times V \times 3$. Finally, the unpatchified voxel tensor is devoxelized into a $N \times 3$ tensor as the output noise $\epsilon_\theta(\mathbf{x}_t, t)$, matching with the ground truth Gaussian noise $\epsilon$ in the point cloud space.

**Model Scaling.** Our DiT-3D is designed to be scalable, adapting to varying voxel sizes, patch sizes, and model sizes. Specifically, it can flexibly accommodate voxel dimensions of 16, 32, 64, patch dimensions of 2, 4, 8, and model complexity ranging from Small, Base, Large and Extra Large, as demonstrated in DiT [1]. For instance, a model designated as DiT-3D-S/4 refers that it utilizes the Small configuration of the DiT model [1], with a patch size $p$ of 4.

## 3.3 Efficient Modality/Domain Transfer with Parameter-efficient Fine-tuning

Leveraging the scalability of the plain diffusion transformer, we investigate parameter-efficient fine-tuning for achieving modality and domain transferability. To facilitate modality transfer from 2D to 3D, we can leverage the knowledge pre-trained on large-scale 2D images using DiT [1]. For domain transfer from a source class to target classes, we train DiT-3D on a single class (*e.g.* chair) and transfer the model's parameters to other classes (*e.g.* airplane, car).

**Modality Transfer: 2D (ImageNet) → 3D (ShapeNet).** As large-scale pre-trained DiT checkpoints[2] are readily available, we can skip training our diffusion transformer from scratch. Instead, we can load most of the weights from the DiT [1] pre-trained on ImageNet [37] into our DiT-3D and continue with fine-tuning. To further optimize training efficiency, we adopt the parameter-efficient fine-tuning approach described in recent work, DiffFit [4], which involves freezing the majority of parameters and only training the newly-added scale factors, bias term, normalization, and class condition modules. It's worth noting that we initialize $\gamma$ to 1, which is then multiplied with the frozen layers.

**Domain Transfer: Source Class → Target Class.** Given a pre-trained DiT-3D model on chair data, we can use the parameter-efficient fine-tuning approach to extend its applicability to new categories. Specifically, following the same methodology as described above, we leverage the fine-tuning strategy of DiffFit and obtain satisfactory generation results.

---

[2]https://github.com/facebookresearch/DiT/tree/main/diffusion

Table 1: Comparison results (%) on shape metrics of our DiT-3D and baseline models.

| Method | Chair | | | | Airplane | | | | Car | | | |
|---|---|---|---|---|---|---|---|---|---|---|---|---|
| | 1-NNA (↓) | | COV (↑) | | 1-NNA (↓) | | COV (↑) | | 1-NNA (↓) | | COV (↑) | |
| | CD | EMD | CD | EMD | CD | EMD | CD | EMD | CD | EMD | CD | EMD |
| r-GAN [8] | 83.69 | 99.70 | 24.27 | 15.13 | 98.40 | 96.79 | 30.12 | 14.32 | 94.46 | 99.01 | 19.03 | 6.539 |
| l-GAN (CD) [8] | 68.58 | 83.84 | 41.99 | 29.31 | 87.30 | 93.95 | 38.52 | 21.23 | 66.49 | 88.78 | 38.92 | 23.58 |
| l-GAN (EMD) [8] | 71.90 | 64.65 | 38.07 | 44.86 | 89.49 | 76.91 | 38.27 | 38.52 | 71.16 | 66.19 | 37.78 | 45.17 |
| PointFlow [9] | 62.84 | 60.57 | 42.90 | 50.00 | 75.68 | 70.74 | 47.90 | 46.41 | 58.10 | 56.25 | 46.88 | 50.00 |
| SoftFlow [10] | 59.21 | 60.05 | 41.39 | 47.43 | 76.05 | 65.80 | 46.91 | 47.90 | 64.77 | 60.09 | 42.90 | 44.60 |
| SetVAE [18] | 58.84 | 60.57 | 46.83 | 44.26 | 76.54 | 67.65 | 43.70 | 48.40 | 59.94 | 59.94 | 49.15 | 46.59 |
| DPF-Net [11] | 62.00 | 58.53 | 44.71 | 48.79 | 75.18 | 65.55 | 46.17 | 48.89 | 62.35 | 54.48 | 45.74 | 49.43 |
| DPM [29] | 60.05 | 74.77 | 44.86 | 35.50 | 76.42 | 86.91 | 48.64 | 33.83 | 68.89 | 79.97 | 44.03 | 34.94 |
| PVD [12] | 57.09 | 60.87 | 36.68 | 49.24 | 73.82 | 64.81 | 48.88 | 52.09 | 54.55 | 53.83 | 41.19 | 50.56 |
| LION [13] | 53.70 | 52.34 | 48.94 | 52.11 | 67.41 | 61.23 | 47.16 | 49.63 | 53.41 | 51.14 | **50.00** | **56.53** |
| GET3D [14] | 75.26 | 72.49 | 43.36 | 42.77 | – | – | – | – | 75.26 | 72.49 | 15.04 | 18.38 |
| MeshDiffusion [15] | 53.69 | 57.63 | 46.00 | 46.71 | 66.44 | 76.26 | 47.34 | 42.15 | 81.43 | 87.84 | 34.07 | 25.85 |
| DiT-3D (ours) | **49.11** | **50.73** | **52.45** | **54.32** | **62.35** | **58.67** | **53.16** | **54.39** | **48.24** | **49.35** | **50.00** | 56.38 |

## 3.4 Relationship to DiT [1]

Our DiT-3D contains multiple different and efficient designs for 3D shape generation compared with DiT [1] on 2D image generation:

- We effectively achieve the diffusion space on voxelized point clouds, while DiT needs the latent codes from a pre-trained variational autoencoder as the denoising target.

- Our plain diffusion transformer first incorporates frequency-based sine-cosine 3D positional embeddings with patch embeddings for voxel structure locality.

- We are the first to propose efficient 3D window attention in the transformer blocks for reducing the complexity of the self-attention operator in DiT.

- We add a devoxelized operator to the final output of the last linear layer from DiT for denoising the noise prediction in the point cloud space.

## 4 Experiments

### 4.1 Experimental Setup

**Datasets.** Following most previous works [12, 13], we use ShapeNet [38] Chair, Airplane, and Car as our primary datasets for 3D shape generation. For each 3D shape, we sample 2,048 points from 5,000 provided points in [38] for training and testing. We also use the same dataset splits and pre-processing in PointFlow [9], which normalizes the data globally across the whole dataset.

**Evaluation Metrics.** For comprehensive comparisons, we follow prior work [12, 13] and use Chamfer Distance (CD) and Earth Mover's Distance (EMD) as our distance metrics in computing 1-Nearest Neighbor Accuracy (1-NNA) and Coverage (COV) as main metrics to measure generative quality. 1-NNA calculates the leave-one-out accuracy of the 1-NN classifier to quantify point cloud generation performance, which is robust and correlates with generation quality and diversity. A lower 1-NNA score is better. COV measures the number of reference point clouds matched to at least one generated shape, correlating with generation diversity. Note that a higher COV score is better but does not measure the quality of the generated point clouds since low-quality but diverse generated point clouds can achieve high COV scores.

**Implementation.** Our implementation is based on the PyTorch [39] framework. The input voxel size is $32 \times 32 \times 32 \times 3$, *i.e.*, $V = 32$. The final linear layer is initialized with zeros, and other weights initialization follows standard techniques in ViT [35]. The models were trained for 10,000 epochs using the Adam optimizer [40] with a learning rate of $1e - 4$ and a batch size of 128. We set $T = 1000$ for experiments. In the default setting, we use S/4 with patch size $p = 4$ as the backbone. Note that we utilize 3D window attention in partial blocks (*i.e.*, 0,3,6,9) and global attention in other blocks.

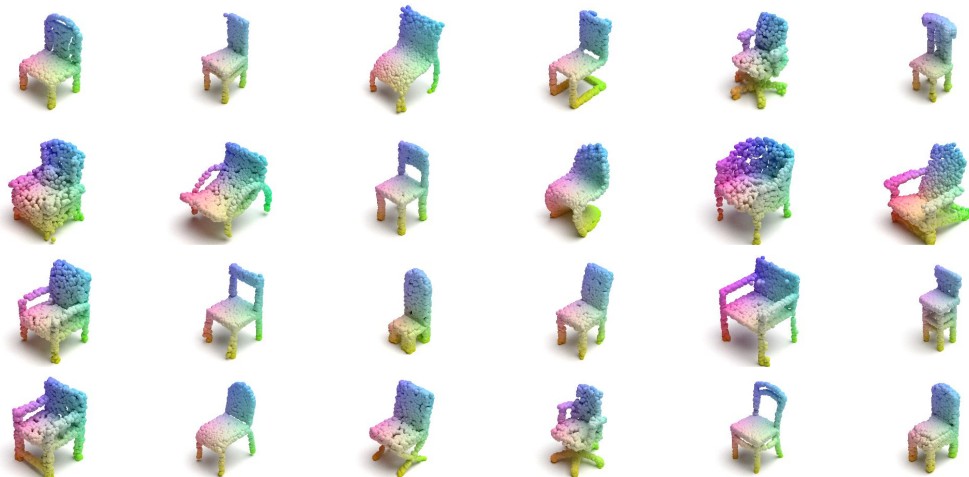

Figure 3: Qualitative visualizations of high-fidelity and diverse 3D point cloud generation.

## 4.2 Comparison to State-of-the-art Works

In this work, we propose a novel and effective diffusion transformer for 3D shape generation. In order to validate the effectiveness of the proposed DiT-3D, we comprehensively compare it to previous non-DDPM and DDPM baselines. 1) r-GAN, l-GAN [8]: (2018'ICML): generative models based on GANs trained on point clouds (l-GAN) and latent variables (l-GAN); 2) PointFlow [9] (2019'ICCV): a probabilistic framework to generate 3D point clouds from a two-level hierarchy of distributions with the continuous normalizing flow; 3) SoftFlow [10] (2020'NeurIPS): a probabilistic framework for training normalizing flows on manifolds to estimate the distribution of various shapes; 4) SetVAE [18] (2021'CVPR): a hierarchical variational autoencoder for sets to learn latent variables for coarse-to-fine dependency and permutation invariance; 5) DPF-Net [11] (2020'ECCV): a discrete latent variable network that builds on normalizing flows with affine coupling layers; 6) DPM [29] (2021'ICCV): the first DDPM approach to learn the reverse diffusion process for point clouds as a Markov chain conditioned on shape latent; 7) PVD [12] (2021'ICCV): a strong DDPM baseline based on the point-voxel representation of 3D shapes; 8) LION [13] (2022'NeurIPS): a recent method based on two hierarchical DDPMs in global latent and latent points spaces; 9) GET3D [14] (2022'NeurIPS): a generative model that directly generates explicit textured 3D meshes based on two latent codes (a 3D SDF and a texture field); 10) MeshDiffusion [15] (2023'ICLR): a very recent DDPM method using graph structure of meshes and deformable tetrahedral grid parametrization of 3D mesh shapes.

For chair generation, we report the quantitative comparison results in Table 1. As can be seen, we achieved the best performance in terms of all metrics compared to previous non-DDPM and DDPM baselines. In particular, the proposed DiT-3D significantly outperforms DPF-Net [11], the current state-of-the-art normalizing flows baseline, decreasing by 12.89 1-NNA@CD & 7.80 1-NNA@EMD, and increasing by 7.74 COV@CD & 3.8 COV@EMD. Moreover, we achieve superior performance gains compared to MeshDiffusion [15], the current state-of-the-art DDPM baseline on meshes, which implies the importance of replacing the U-Net with a plain diffusion transformer from observed point clouds for generating high-fidelity 3D shapes. Meanwhile, our DiT-3D outperforms LION [15] by a large margin, where we achieve the performance gains of 4.59 1-NNA@CD & 1.61 1-NNA@EMD, and 3.51 COV@CD & 2.21 COV@EMD. These significant improvements demonstrate the superiority of our method in 3D shape generation. In addition, significant gains in airplane and car generations can be observed in Table 1. These qualitative results also showcase the effectiveness of applying a plain diffusion transformer to operate the denoising process from point clouds for generating high-fidelity and diverse shapes, as shown in Figure 3.

## 4.3 Experimental Analysis

In this section, we performed ablation studies to demonstrate the benefit of introducing three main 3D design components (voxel diffusion, 3D positional embeddings, and 3D window attention) in 3D shape generation. We also conducted extensive experiments to explore the efficiency of 3D window attention, modality and domain transferability, and scalability.

Table 2: **Ablation studies** on 3D adaptation components of our DiT-3D.

| Voxel Diffusion | 3D Pos Embed | 3D Window Attention | Training Cost (hours) | 1-NNA (↓) CD | EMD | COV (↑) CD | EMD |
|---|---|---|---|---|---|---|---|
| ✗ | ✗ | ✗ | 86.53 | 99.86 | 99.93 | 7.768 | 4.653 |
| ✓ | ✗ | ✗ | 91.85 | 67.46 | 69.47 | 38.97 | 41.74 |
| ✓ | ✓ | ✗ | 91.85 | 51.99 | **49.94** | **54.76** | **57.37** |
| ✓ | ✓ | ✓ | **41.67** | **49.11** | 50.73 | 52.45 | 54.32 |

Table 3: **Transferability studies** on modality and domain with parameter-efficient fine-tuning.

| ImageNet Pre-train | Efficient Fine-tuning | Params (MB) | 1-NNA (↓) CD | EMD | COV (↑) CD | EMD |
|---|---|---|---|---|---|---|
| ✗ | ✗ | 32.8 | 51.99 | 49.94 | **54.76** | **57.37** |
| ✓ | ✗ | 32.8 | **49.07** | **49.76** | 53.26 | 55.75 |
| ✓ | ✓ | **0.09** | 50.87 | 50.23 | 52.59 | 55.36 |

(a) Modality transfer.

| Source Domain | Target Domain | Params (MB) | 1-NNA (↓) CD | EMD | COV (↑) CD | EMD |
|---|---|---|---|---|---|---|
| Chair | Chair | 32.8 | **51.99** | **49.94** | **54.76** | **57.37** |
| Airplane | Chair | **0.09** | 52.56 | 50.75 | 53.71 | 56.32 |
| Airplane | Airplane | 32.8 | **62.81** | **58.31** | **55.04** | **54.58** |
| Chair | Airplane | **0.09** | 63.58 | 59.17 | 53.25 | 53.68 |

(b) Domain transfer.

**Ablation on 3D Design Components.** In order to validate the effectiveness of the introduced 3D adaptation components (voxel diffusion, 3D positional embeddings, and 3D window attention), we ablate the necessity of each module and report the quantitative results in Table 2. Note that no voxel diffusion means we directly perform the denoising process on point coordinates without voxelized point clouds and devoxelization prediction. We can observe that adding bearable voxel diffusion to the vanilla baseline highly decreases the results of 1-NNA (by 32.40 @CD and 30.46 @AUC) and increase the performance of COV (by 31.202 @CD and 37.087 @EMD), which demonstrates the benefit of voxelized point clouds and devoxelization prediction in denoising process for 3D shape generation. Meanwhile, introducing 3D positional embedding in the baseline with voxel diffusion also increases the shape generation performance in terms of all metrics. More importantly, incorporating 3D window attention and two previous modules together into the baseline significantly decreases the training cost by 44.86 hours and results of 1-NNA by 50.75 @CD and 49.2 @EMD, and raises the performance of COV by 44.682 @CD and 49.667 @EMD. These improving results validate the importance of the proposed 3D adaptation components in the plain diffusion transformer to operate the denoising process from observed point clouds for 3D shape generation.

**Influence of 2D Pretrain (ImageNet).** In order to show the modality transferability of the proposed approach from 2D ImageNet pre-trained weights to 3D generation with parameter-efficient fine-tuning, we report the ablation results of ImageNet pre-train and efficient fine-tuning on chair generation in Table 3a. From comparisons, two main observations can be derived: 1) With the initialization with 2D ImageNet pre-trained weights, the proposed DiT-3D improves the quality of shape generation by decreasing 1-NNA by 2.92@CD and 0.18@EMD. 2) Incorporating parameter-efficient fine-tuning into 2D ImageNet pre-trained weights highly decreases the training parameters while achieving competitive generation performance.

**Transferability in Domain.** In addition, we explore the parameter-efficient fine-tuning for domain transferability in Table 3b. By only training 0.09MB parameters of models from the source class to the target class, we can achieve a comparable performance of quality and diversity in terms of all metrics. These results indicate that our DiT-3D can support flexible transferability on modality and domain, which differs from previous 3D generation methods [12, 13] based on U-Net as the backbone of DDPMs.

**Scaling Patch size, Voxel size and Model Size.** To explore the scalability of our plain diffusion transformer to flexible designs, we ablate the patch size from $\{2, 4, 8\}$, voxel size from $\{16, 32, 64\}$, and the model size from $\{S/4, B/4, L/4, XL/4\}$. As seen in Table 4a, when the patch size is 2, the proposed DiT-3D achieves the best performance. This trend is also observed in the original DiT [1] work for 2D image generation. In addition, increasing the voxel size from 16 to 64 for the input of the diffusion denoising process raises the performance in terms of all metrics, as shown in Table 4b. More importantly, we can still observe performance gains by scaling up the proposed plain diffusion transformer to XL/4 when the model is trained for 2,000 epochs. These promising results further demonstrate the strong scalability of our DiT-3D to flexible patch size, voxel size, and model sizes for generating high-fidelity 3D shapes.

Table 4: **Scalability studies** on flexible patch, voxel, and model sizes.

| Patch Size | 1-NNA (↓) | | COV (↑) | |
|---|---|---|---|---|
| | CD | EMD | CD | EMD |
| 8 | 53.84 | 51.20 | 50.01 | 52.49 |
| 4 | 51.99 | 49.94 | **54.76** | **57.37** |
| 2 | **51.78** | **49.69** | 54.54 | 55.94 |

(a) Patch size.

| Voxel Size | 1-NNA (↓) | | COV (↑) | |
|---|---|---|---|---|
| | CD | EMD | CD | EMD |
| 16 | 54.00 | 50.60 | 50.73 | 52.26 |
| 32 | 51.99 | 49.94 | 54.76 | 57.37 |
| 64 | **50.32** | **49.73** | **55.45** | **57.32** |

(b) Voxel size.

| Model Size | Params (MB) | 1-NNA (↓) | | COV (↑) | |
|---|---|---|---|---|---|
| | | CD | EMD | CD | EMD |
| S/4 | 32.8 | 56.31 | 55.82 | 47.21 | 50.75 |
| B/4 | 130.2 | 55.59 | 54.91 | 50.09 | 52.80 |
| L/4 | 579.0 | 52.96 | 53.57 | 51.88 | 54.41 |
| XL/4 | 674.7 | **51.95** | **52.50** | **52.71** | **54.31** |

(c) Model size.

## 5 Conclusion

In this work, we present DiT-3D, a novel plain diffusion transformer for 3D shape generation, which can directly operate the denoising process on voxelized point clouds. Compared to existing U-Net approaches, our DiT-3D is more scalable in model size and produces much higher quality generations. Specifically, we incorporate 3D positional and patch embeddings to aggregate input from voxelized point clouds. We then incorporate 3D window attention into Transformer blocks to reduce the computational cost of 3D Transformers, which can be significantly high due to the increased token length resulting from the additional dimension in 3D. Finally, we leverage linear and devoxelization layers to predict the denoised point clouds. Due to the scalability of the Transformer, DiT-3D can easily support parameter-efficient fine-tuning with modality and domain transferability. Empirical results demonstrate the state-of-the-art performance of the proposed DiT-3D in high-fidelity and diverse 3D point cloud generation.

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
