# DiT-3D: Exploring Plain Diffusion Transformers for 3D Shape Generation
## *(Supplementary Material)*

**Shentong Mo**[1]  **Enze Xie**[2][*]  **Ruihang Chu**[3]  **Lewei Yao**[2]
**Lanqing Hong**[2]  **Matthias Nießner**[4]  **Zhenguo Li**[2]

[1]MBZUAI, [2]Huawei Noah's Ark Lab, [3]CUHK, [4]TUM
https://DiT-3D.github.io

In this appendix, we first provide additional experimental analyses on multi-class training and DDPM sampling steps in Section A. Furthermore, we showcase qualitative visualizations for comparisons with state-of-the-art works, visualizations of the diffusion process, and more high-fidelity visualizations in Section B. Finally, we thoroughly discuss this work's limitation and broader impact in Section C.

## A  Additional Experimental Analyses

### A.1  Results on Multi-class Training

In order to show the effectiveness of our DiT-3D on multi-class training, we change the training classes from {Chair}, {Chair,Car}, {Chair,Car,Airplane} and test on chair class in Table 1. We can observe that the proposed diffusion transformer achieves competitive generation results against category-specific models for all metrics by using learnable class embeddings as the condition after multi-class training. This benefits us in training only one global model for all classes simultaneously instead of training class-specific models multiple times, which differs from previous DDPM-based approaches without class embeddings involved.

Table 1: **Exploration studies** on multi-class training. One global model for all three classes achieves competitive results against category-specific models trained on only one class.

| Train Class | Test Class | 1-NNA ($\downarrow$) | | COV ($\uparrow$) | |
| --- | --- | --- | --- | --- | --- |
| | | CD | EMD | CD | EMD |
| Chair | Chair | **51.99** | **49.94** | **54.76** | **57.37** |
| Chair, Car | Chair | 52.68 | 50.62 | 54.15 | 56.83 |
| Chair, Car, Airplane | Chair | 53.35 | 51.84 | 52.81 | 55.30 |

### A.2  Effect of Sampling Steps

Furthermore, we explore the effect of DDPM sampling steps $T$ on the final performance during the inference stage in Figure 1. As can be seen, the proposed DiT-3D achieves the best results (lowest 1-NNA and highest COV) for all metrics (CD and EMD) when the number of sampling steps is set to 1000. This trend is consistent with similar conclusions in the prior DDPM work [1].

---

[*]Corresponding author.

37th Conference on Neural Information Processing Systems (NeurIPS 2023).

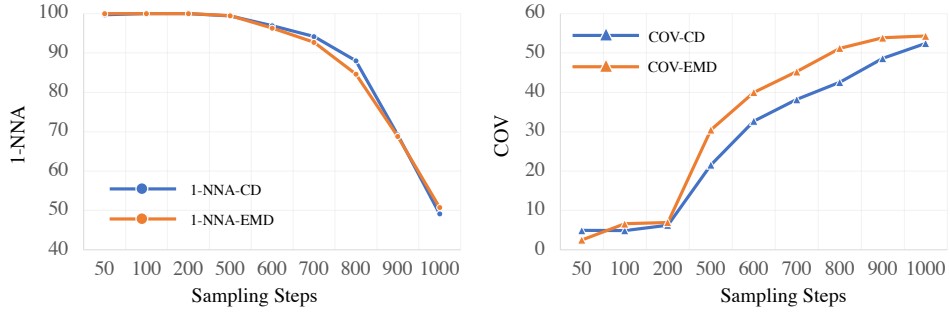

Figure 1: Effect of sampling steps on 3D shape generation (*Chair*) during the inference stage.

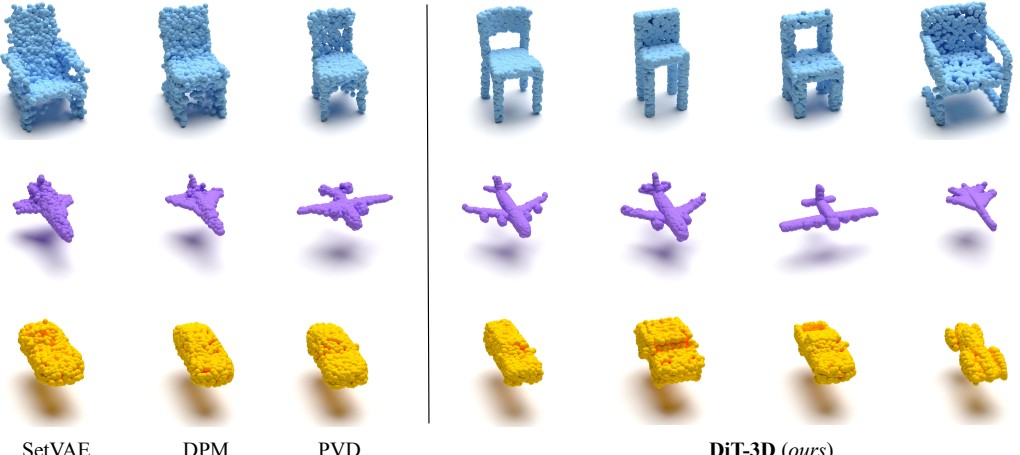

| SetVAE | DPM | PVD | **DiT-3D** (*ours*) |

Figure 2: Qualitative comparisons with state-of-the-art works. The proposed DiT-3D generates high-fidelity and diverse point clouds of 3D shapes for each category.

## B   Qualitative Visualizations

### B.1   Comparisons with State-of-the-art Works

In order to qualitatively evaluate the generated 3D shapes, we compare the proposed DiT-3D with SetVAE [2], DPM [3], and PVD [4] on generated 3D point clouds of all three class in Figure 2. From comparisons, we can observe that the qualities of 3D point clouds generated by our framework are superior to SetVAE [2], a hierarchical variational autoencoder for sets to learn latent variables for coarse-to-fine dependency and permutation invariance. Meanwhile, we achieve much better results than DPM [3], the first diffusion denoising probabilistic model on point cloud generation. More importantly, the proposed DiT-3D achieves high-fidelity and diverse results compared to the strong diffusion model based on point voxels, PVD [4]. These visualizations further showcase the superiority of our DiT-3D in generating high-fidelity and diverse shapes by using a plain diffusion transformer to operate the denoising process from point clouds.

### B.2   Visualizations of Diffusion Process

Furthermore, we visualize the diffusion process of generated *Chair* shapes from 1000 sampling steps in Figure 3, generating 6 shapes from random noise to the final 3D shapes for each sample. From left to right, we can observe that our DiT-3D achieves a meaningful diffusion process to produce high-fidelity and diverse shapes. When the number of sampling steps is closer to 1000, the generated shapes are more realistic, while they are more like random noises in the initial few sampling steps. These qualitative diffusion process results also showcase the effectiveness of applying a plain diffusion transformer to generate high-fidelity and diverse shapes. The results of the diffusion process for *Airplane* and *Car* shapes generated from 1000 sampling steps are reported in Figure 4 and 5.

### B.3 More Visualizations of Generated Shapes

To qualitatively showcase the high-fidelity and diverse properties of generated shapes, we visualize more generated samples from all three classes in Figure 6, 7, and 8. These qualitative visualizations demonstrate the effectiveness of the proposed 3D design components in a plain diffusion transformer to produce high-fidelity and diverse shapes by achieving the denoising process from point clouds of three categories directly.

## C  Discussion

**Limitation & Future Work.** This work thoroughly explores the plain diffusion transformer on point clouds for generating high-fidelity and diverse 3D shapes. However, we have yet to explore the potential of other 3D modalities, such as signed distance fields (SDFs) and meshes, or scaling our DiT-3D to large-scale training on more 3D shapes. These directions are promising, and we will leave them as the future work.

**Broader Impact.** The proposed DiT-3D generates high-fidelity and diverse 3D shapes from training samples in the existing ShapeNet benchmark, which might cause the model to learn internal biases in the data. These biased problems should be carefully solved for the deployment of real applications.

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

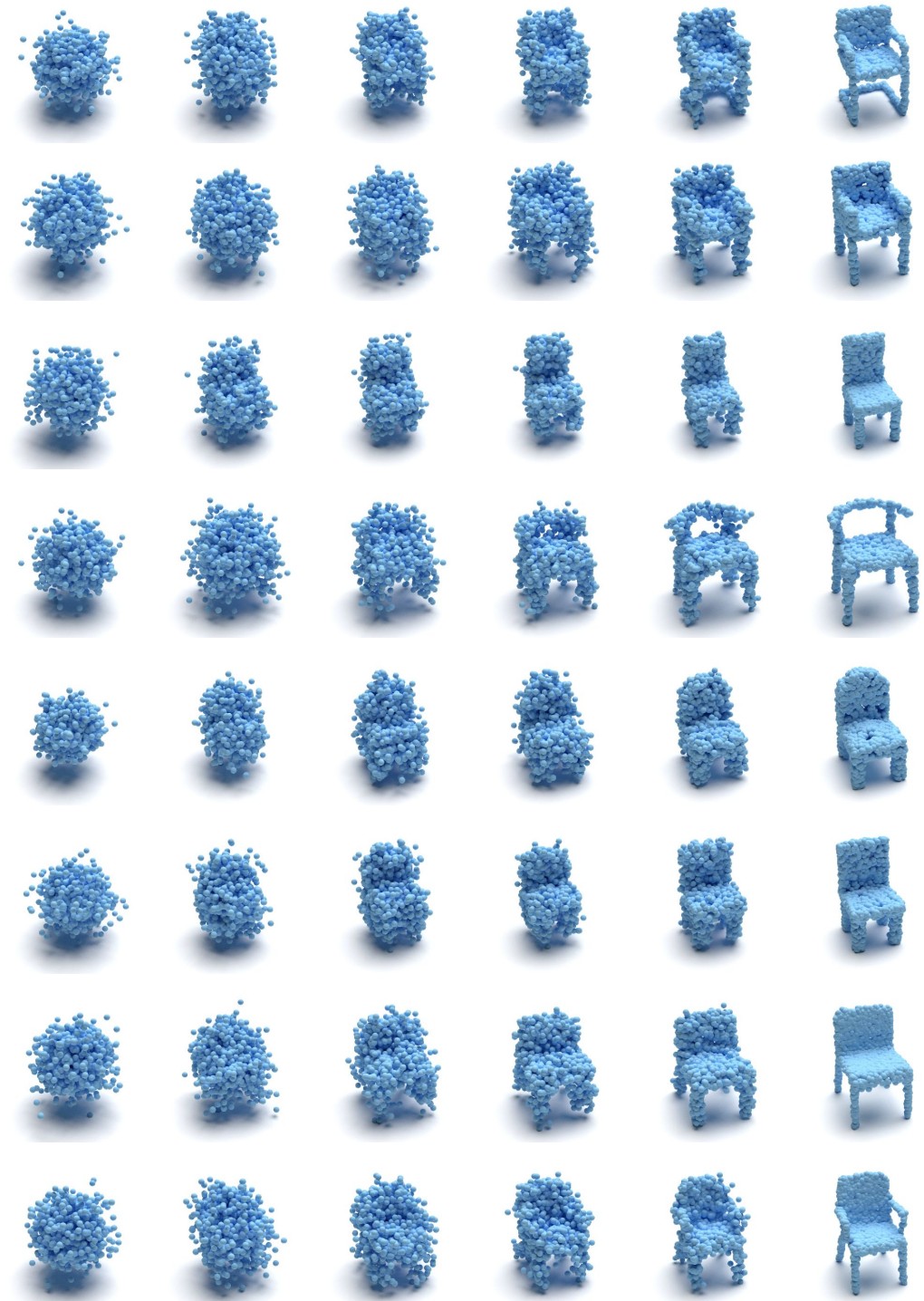

Figure 3: Qualitative visualizations of the diffusion process on *Chair* shape generation. The results of generating from random noise to final 3D shapes are shown in left-to-right order.

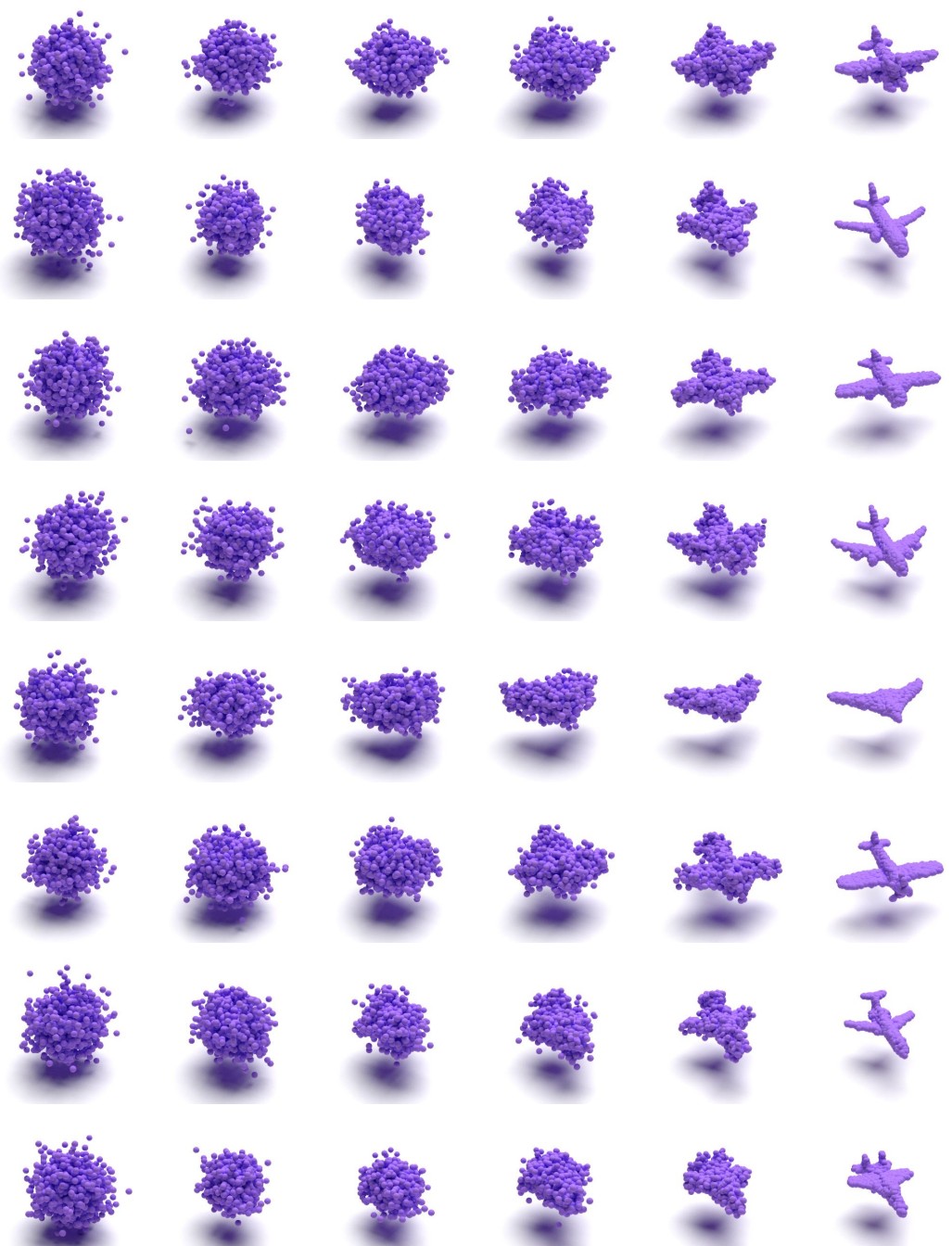

Figure 4: Qualitative visualizations of the diffusion process on *Airplane* shape generation. The results of generating from random noise to final 3D shapes are shown in left-to-right order.

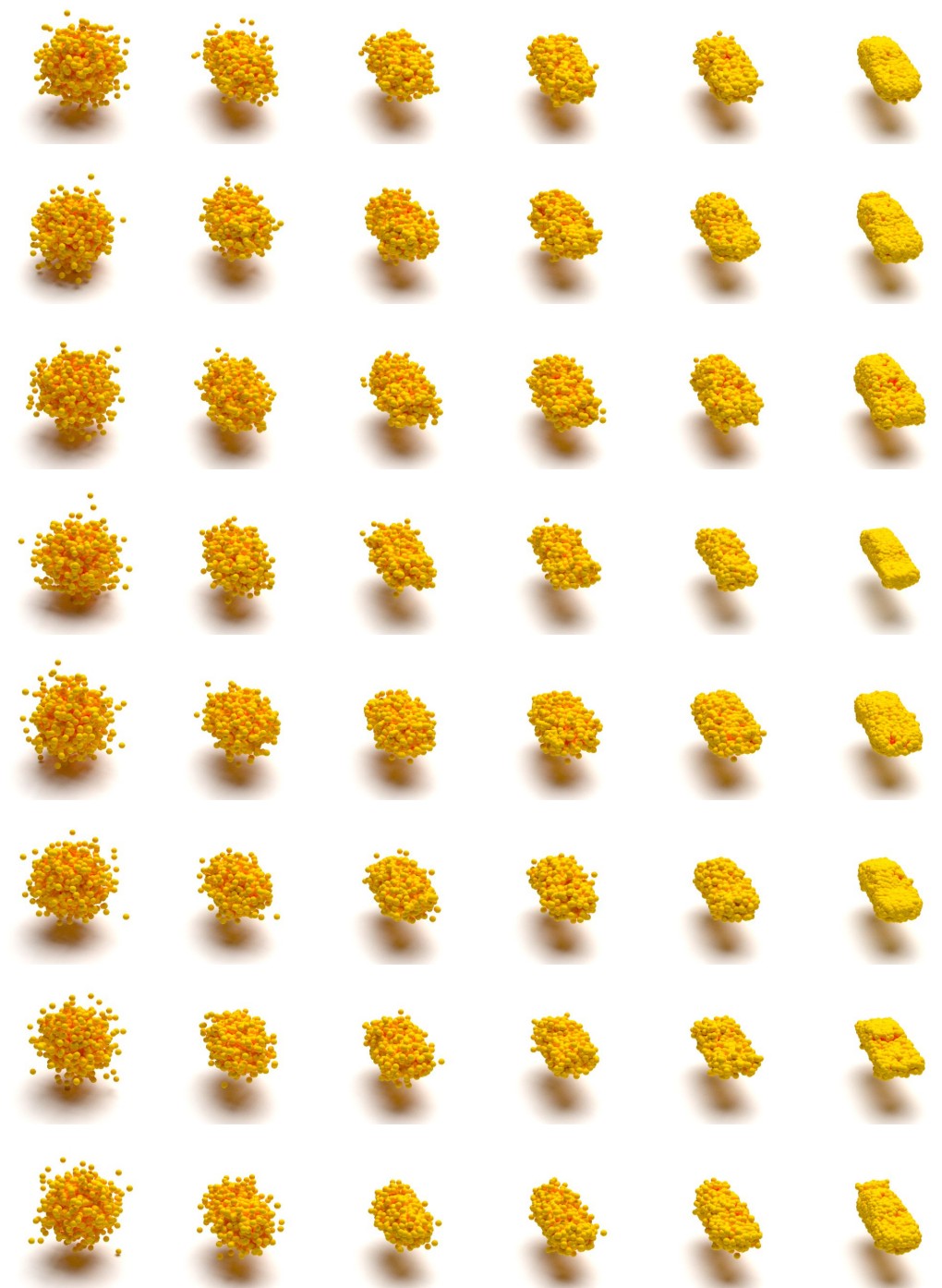

Figure 5: Qualitative visualizations of the diffusion process on *Car* shape generation. The results of generating from random noise to final 3D shapes are shown in left-to-right order.

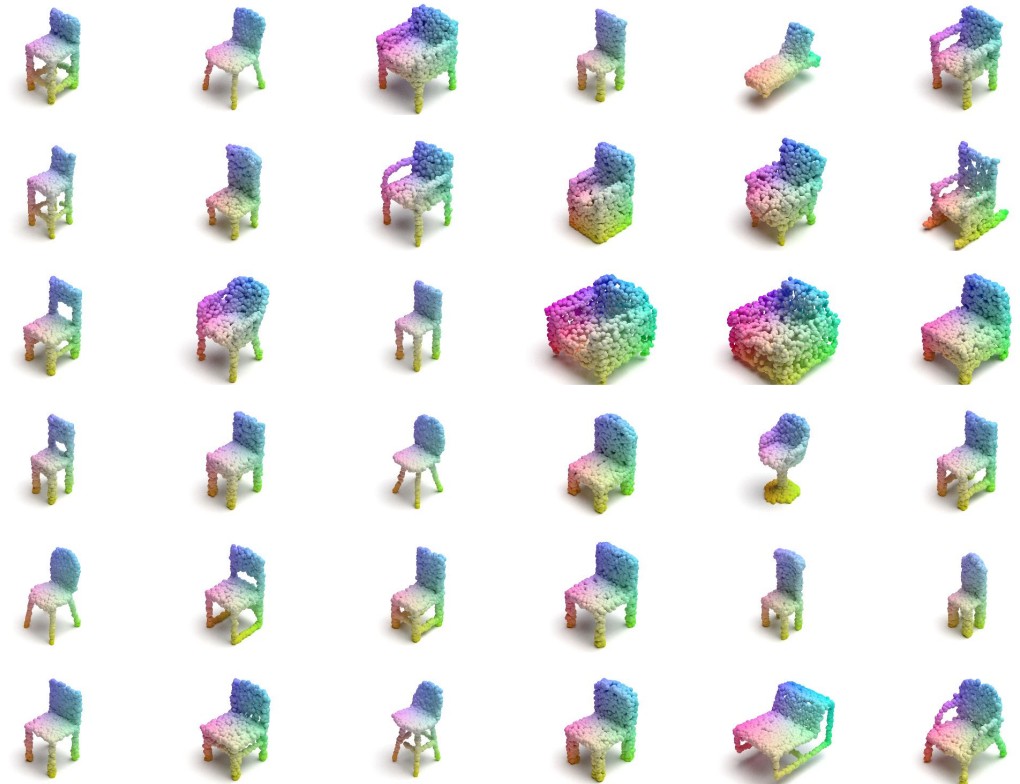

Figure 6: Qualitative visualizations of high-fidelity and diverse results on *Chair* shape generation.

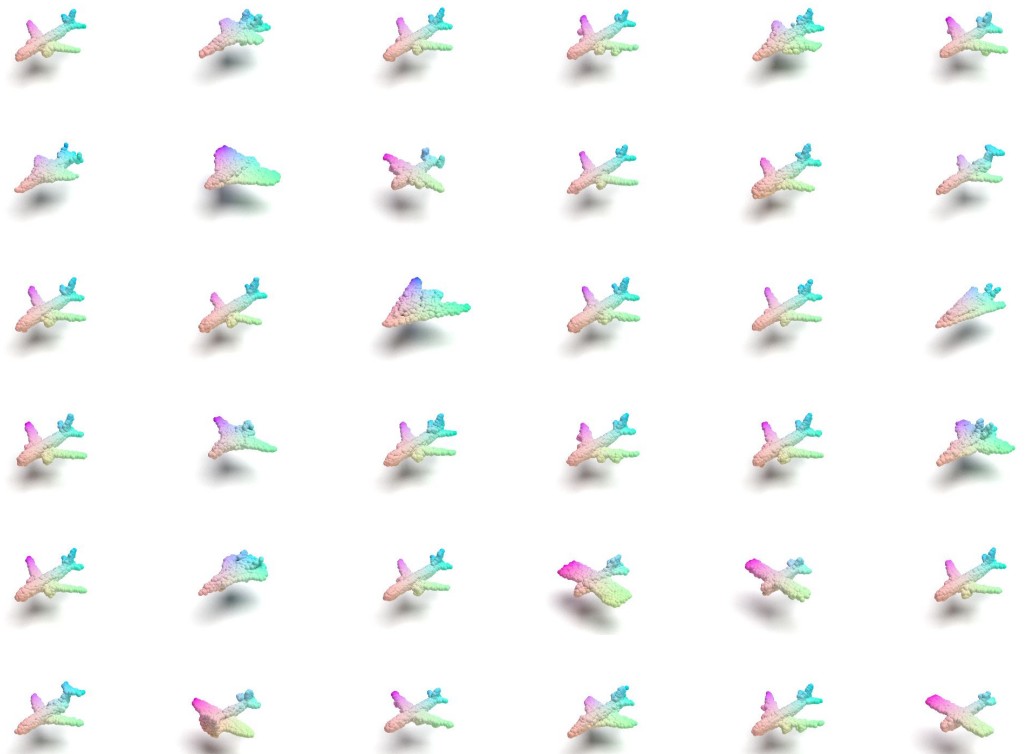

Figure 7: Qualitative visualizations of high-fidelity and diverse results on *Airplane* shape generation.

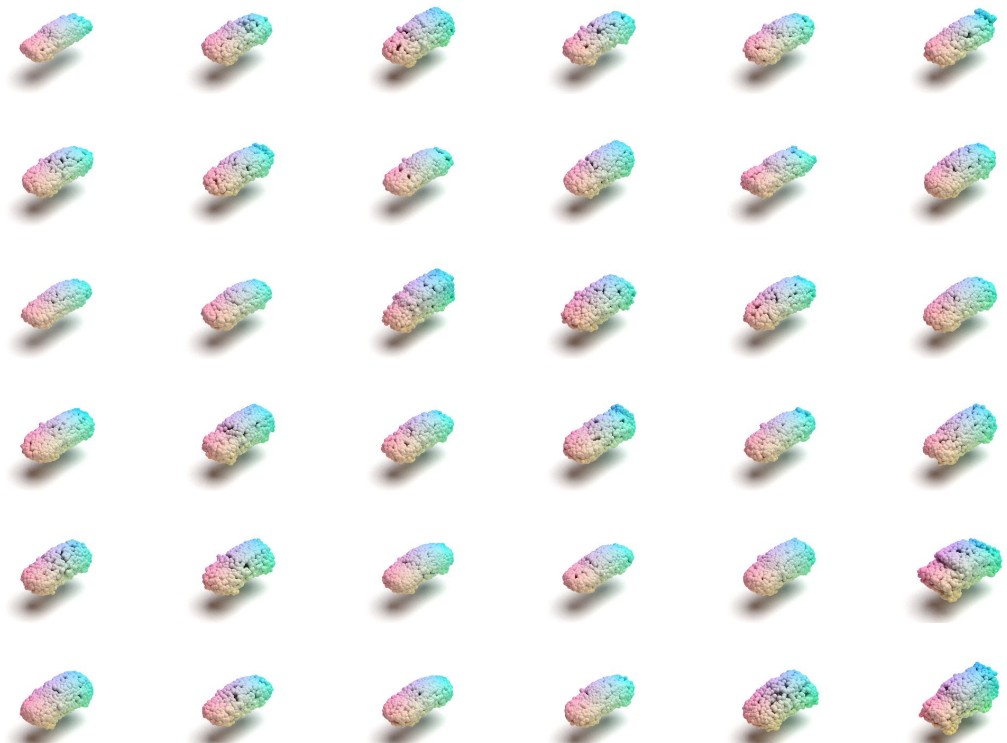

Figure 8: Qualitative visualizations of high-fidelity and diverse results on *Car* shape generation.