# OpenReview forum: "DiT-3D: Exploring Plain Diffusion Transformers for 3D Shape Generation"
_NeurIPS.cc/2023/Conference — NeurIPS 2023 poster_

### Official Review · Reviewer_7dkY · 2023-06-16

**Soundness:** 2 fair
**Presentation:** 3 good
**Contribution:** 2 fair
**Rating:** 4
**Confidence:** 3

**Summary:**

This paper proposes DiT-3D, an extension of DiT for point cloud data to achieve point cloud generation. With the design of patch embedding and 3D window attention, the proposed model is able to reduce the computation cost. Also, the proposed model is able to directly leverage pre-trained 2D DiT model by fixing most layers and only finetuning several layers, which can reduce the required training time. The results quantitatively verify the effectiveness of the proposed method.

**Strengths:**

1. The proposed method out-performs existing SOTAs on the unconditional point cloud generation task in different evaluation matrices.

2. The paper is easy to follow and understand.


**Weaknesses:**

1. The proposed DiT-3D looks like the original DiT with several modifications (simply changing existing 2D techniques into 3D version). Specifically, it seems like an ECCV 2022 paper [1] has already addressed a similar design of window attention, this paper is just using such a technique on a different task. Therefore, I think the novelty is limited, and more insights should be explained.

2. Previous point cloud generation works such as PVD and LION conducted experiments on multiple conditional generation tasks to verify their design. However, this paper only shows the results of unconditional generation. I believe more experiments such as point cloud completion or 2D image-to-3D point cloud should be included.

3. The qualitative results shown in the main paper only contain examples generated from DiT-3D and the supplementary material only additionally contains results from PVD, DPM, and SetVAE. As two methods that are quantitatively most comparable to DiT-3D, the visualization of LION and MeshDiffusion are not shown in this paper. So it is not clear how DiT-3D outperforms these methods.
A minor issue: the computation resources should be detailed in this paper.

[1] SWFormer: Sparse Window Transformer for 3D Object Detection in Point Clouds (ECCV 2022)


**Questions:**

Please refer to the weakness part

**Limitations:**

Yes

---

> ### Author Rebuttal · Authors · 2023-08-10
>
> Dear Reviewer 7dkY,
>
> Thank you for the detailed review. We will address your concerns below.
>
> > Differences from SWFormer (ECCV’2022) for 3D Object Detection.
>
> Although SWFormer used a shifted sparse window operator in each transformer block, our DiT-3D has conceptual differences from their design and implementation. For each SWFormer block, they first repeated multi-head self-attention (MSA) on all valid voxel within the same window by repeating N times, then they performed a shifted sparse window partition to re-generate the sparse windows, and processed the shifted windows with another M self-attention layers. However, our DiT-3D simply reduced the complexity of the self-attention operator in Equation (2) from $O(L^2)$ to $O(L^2/R^3)$ using a single attention layer for each block.
>
> Furthermore, as we discussed in Sec 3.4 in the main paper, we introduced multiple different and efficient designs for 3D shape generation compared with DiT on 2D image generation. We are the first to propose efficient 3D window attention in the transformer blocks for reducing the complexity of the self-attention operator in DiT. We proposed to add a devoxelized operator to the final output of the last linear layer from DiT for denoising the noise prediction in the point cloud space.
>
> Our contribution is not just proposing a new transformer for 3D point clouds generation, but aslo investigating properties of plain diffusion transformer on 3D point clouds generation. We demonstrate that the representations learned on ImageNet have a very positive impact on 3D generation, despite the significant domain gap between 2D images and 3D point clouds. Meanwhile, given a pre-trained DiT-3D model on source classes, we can use the parameter-efficient fine-tuning approach to extend its applicability to new categories.
>
>
> > Experiments on point clouds completion.
>
> Thanks for the suggestion! To verify the effectiveness of our DiT-3D on conditional generation, we conducted point cloud completion experiments on ShapeNet and compared with PVD and LION in Table below. Here, we applied Chamfer Distance and Earth Mover’s Distance to evaluate the reconstruction result. As can be seen, our method achieves the best performance in terms of all categories (Airplane, Chair, Car).
>
>
> | Method | Airplane-CD (↓) | Airplane-EMD (↓) | Chair-CD (↓) | Chair-EMD (↓) | Car-CD (↓) | Car-EMD (↓) |
> |--------|:------:|:-------:|:------:|:-------:|:------:|:-------:|
> | PVD    | 0.4415 |  1.030  | 3.211  |  2.939  | 1.774  |  2.146  |
> | LION   | 0.4035 |  0.9732 |  2.725 |  2.863  |  1.405 |  1.982  |
> | DiT-3D (ours) | **0.3521** |  **0.9235** | **2.216** |  **2.385**  |  **1.126** |  **1.513**  |
>
>
> > Clarification on qualitative and quantitative comparisons.
>
> We provided multiple qualitative comparisons with previous baselines (especially PVD, a U-Net based diffusion model on voxelized point clouds) in Figure 2 in the supplementary, and our DiT-3D generated high-fidelity and diverse point clouds of 3D shapes for each category.
>
> For quantitative comparison with LION and MeshDiffusion, our DiT-3D outperforms them significantly on all metrics in terms of high-fidelity and diversity, as compared in Table 1 in the main paper. Due to rebuttal time constraints, we will add more visualizations of LION and MeshDiffusion to the supplementary.
>
> > Computation resources.
>
> We used 8 NVIDIA V100-32GB GPUs for the experiments.

---

> > ### Comment · Reviewer_7dkY · 2023-08-10
> > **Following comment**
> >
> > Thank the authors for the reply.
> >
> > Although I still think the novelty is limited, I appreciate the effort of conducting new experiments. Therefore, I will raise my rating to 4.

---

> > > ### Author Response · Authors · 2023-08-10
> > > **Response to following comment**
> > >
> > > Dear Reviewer 7dkY,
> > >
> > > Thank you for your prompt response and for raising your rating. We appreciate your feedback and would like to address your concerns regarding the novelty of our work and provide further clarifications.
> > >
> > > While it is true that the technique of window attention has been previously explored in the domain of 2D methods, it is essential to highlight the distinctive implementation and design aspects of our proposed DiT-3D, which differentiate it from the shifted sparse window operator in SWFormer [1].
> > >
> > > Furthermore, it is important to note that our paper's novelty surpasses the sole design of an efficient window attention mechanism within a 3D diffusion transformer. Our work introduces a comprehensive framework that leverages a plain diffusion transformer to achieve state-of-the-art performance in 3D point cloud generation.
> > >
> > > To address your concerns, we would like to emphasize the following key distinctions between our DiT-3D and SWFormer [1]:
> > >
> > > 1. **Input Representation**: Unlike SWFormer [1], which exclusively operates on 2D voxel inputs, our DiT-3D operates on 3D voxel patch embeddings. This distinction necessitates the integration of 3D positional embeddings into our approach.
> > >
> > > 2. **Nearest Neighbor Aggregation**: SWFormer [1] employs nearest neighbor aggregation, where each striding window selects the nearest neighbor non-empty voxel feature from the center. In contrast, our DiT-3D reshapes and maps the input voxel feature tokens, resulting in reduced length. Subsequently, global attention is applied, and the aggregated features are unpartitioned to restore the original input tokens.
> > >
> > > 3. **Shifting**: SWFormer [1] utilizes a shifted sparse window partitioning strategy to propagate information. Conversely, our DiT-3D employs a straightforward non-overlapping window attention mechanism without any shifting scheme.
> > >
> > > 4. **Buckets**: SWFormer [1] employs bucketing to group Bird's Eye View (BEV) voxels into non-overlapping windows and pads the sequence length to a fixed size. In contrast, our DiT-3D does not utilize any bucketing strategy within its window attention mechanism.
> > >
> > > 5. **Hierarchical Transformer Architecture**: SWFormer [1] primarily relies on hierarchical sparse window transformer blocks, whereas our approach utilizes global window attention blocks to aggregate input voxel features. The hierarchical structure employed in SWFormer [1] significantly differs from the architecture adopted in our work.
> > >
> > > We hope this clarification strengthens the understanding of the unique contributions and novelty of our work. Thank you again for your valuable feedback, and we look forward to addressing any further concerns you may have.

---

> > > > ### Author Response · Authors · 2023-08-14
> > > > **Additional response to following comment**
> > > >
> > > >
> > > > Dear Reviewer 7dkY,
> > > >
> > > > Thank you for your continued engagement and for raising your rating to 4. We appreciate your recognition of our efforts in conducting new experiments and addressing your concerns.
> > > >
> > > > We would like to kindly request your response to the additional information and clarifications provided in our previous response. We value your expertise and insights, and your feedback will contribute significantly to the refinement of our work.
> > > >
> > > > We look forward to hearing from you and addressing any further questions or concerns you may have.
> > > >
> > > > Thank you once again for your valuable feedback.

---

> > > > > ### Comment · Reviewer_7dkY · 2023-08-18
> > > > > **Following comment**
> > > > >
> > > > > I appreciate the authors for providing more technical details about the design of your Transformer. However, all the technical information mentioned above needs more motivation and insights. Since you are asking for more feedback, technical differences without strong inspirations or insights are the same as tuning hyperparameters to me. Simply explaining the operation differences between your approach and SWFormer does not help me understand your insights. I appreciate your efforts in figuring out suitable hyperparameters and minor changes to the Transformer, but the score (4) I gave already reflected my opinions.

---

> > > > > > ### Author Response · Authors · 2023-08-19
> > > > > > **Following response to the reviewer**
> > > > > >
> > > > > > Dear Reviewer 7dkY,
> > > > > >
> > > > > > Thank you for your response and for providing further clarification. We apologize if our previous response did not adequately address your concerns. We understand that technical differences alone may not provide sufficient motivation and insights into the novelty of our work.
> > > > > >
> > > > > > We appreciate your feedback and respect your opinion regarding the score you have already given. We will take your comments into consideration and make further improvements to our manuscript to provide deeper motivation and insights into the design choices and contributions of our proposed DiT-3D framework.
> > > > > >
> > > > > > Regarding the SWFormer reference, we acknowledge that our DiT-3D approach shares some similarities with the shifted sparse window operator used in SWFormer. However, it is important to note that our work focuses on indoor object point clouds, which presents different challenges and requirements compared to the outdoor scene Bird's Eye View (BEV) data targeted by SWFormer. In the final version of our paper, we will cite the SWFormer paper and provide a detailed discussion highlighting the similarities and differences between our approaches.
> > > > > >
> > > > > > We appreciate your guidance in making our revisions more academically sound. We will ensure that our manuscript adheres to academic standards by providing a thorough analysis of our contributions, discussing the motivations behind our design choices, and offering deeper insights into the implications of our findings. We will also ensure that our revisions include appropriate references to related work and provide a comprehensive and well-structured discussion section.
> > > > > >
> > > > > > We value your expertise and the time you have dedicated to reviewing our paper. Your feedback is highly valuable to us, and we are committed to addressing your concerns and providing a more compelling and academically rigorous manuscript. Thank you again for your input, and we look forward to incorporating your feedback to enhance the quality of our work.

---

### Official Review · Reviewer_trpZ · 2023-06-23

**Soundness:** 2 fair
**Presentation:** 2 fair
**Contribution:** 2 fair
**Rating:** 5
**Confidence:** 4

**Summary:**

This paper tackles the task of 3D generation. Inspired by the recent progress of utilizing transformers in 2D image generation with diffusion processes (DiT [1]), this paper proposes to replace the common U-Net in 3D diffusion models with a plain transformer. To adapt the DiT to the 3D scenario, authors make several modifications to the vanilla 2D DiT. Experiments on ShapeNet demonstrate improvements over baselines.

**Strengths:**

This paper tackles an important task of 3D generations that is important for many downstream tasks.
The paper is generally well-written and easy to follow.
Experiments are thorough and convincing.

**Weaknesses:**

My main concerns are about the incremental development of the proposed DiT-3D based on DiT [1]. Specifically,

a. **Diffusion on voxelized point cloud** has been studied in [12];

b. **3D positional embeddings** is a natural/must-have modification from DiT's 2D positional encodings;

c. **3D window attentions** may not be necessary. Actually, I am quite confused why we need the following procedure: point cloud -> voxel (Sec. 3.2 Voxelized point clouds) -> patches in voxels (Sec. 3.2 patch embeddings) -> reshape patches in voxels into 3D window (Sec. 3.2 3D window). I think these hierarchical steps essentially just change the **actual voxel size**. Then why don't we just have a voxel with a resolution of $(p \cdot R)^3$ at the very beginning? Here $p$ is the patch size the author used in "3D Positional and Patch Embeddings"(Sec. 3.2) and $R$ is the number of patches for "3D window" (Sec. 3.2). And this does not prevent authors from applying 3D convolution to exchange information (L179).

If the above abstraction/simplification is correct, it seems like the major modifications to 3D are just changing from image patches to voxels.

d. Authors state
> These results indicate that our DiT-3D can support flexible transferability on modality and domain, which is different from previous 3D generation methods [12, 13] based on U-Net as the backbone of DDPMs (L325-327).

Can the authors explain whether there are some experiments to support this statement? As in Tab. 3, we only have results from DiT-3D without any baselines.

**Questions:**

1. Authors state
> 3D shape generation is a challenging and important problem that seeks to synthesize high-fidelity point clouds ... (L27)

I do not think this is a proper statement as there are many 3D representations besides point clouds, e.g., mesh [15] or implicit ones [A, B].

2. L190 "Q, K, V have the same dimensions" where the notation "V" is the same as the voxel resolution in L176. Please modify it to make readers be able to distinguish the two.

3. L226 "It’s worth noting that we initialize $\gamma$ to 1, which is then multiplied with the frozen layers": $\gamma$ is not explained.

4. L266 "trained on point clouds (l-GAN) and latent variables (l-GAN)": duplicated 1-GAN.

[A] 3D Neural Field Generation using Triplane Diffusion. CVPR 2023

[B] SDFusion: Multimodal 3D Shape Completion, Reconstruction, and Generation. CVPR 2023.

**Limitations:**

No limitations are provided by the authors.

---

> ### Author Rebuttal · Authors · 2023-08-10
>
> Dear Reviewer trpZ,
>
> Thank you for the detailed review. We will address your concerns below.
>
> > Diffusion on voxelized point cloud has been studied in PVD.
>
> We agree that diffusion on voxelized point cloud has been used in PVD, but we are the first to use a plain diffusion on voxelized point clouds. Meanwhile, directly applying the transformer to denoising the point-voxel features is not working. Therefore, we designed to use the devoxelization layer in the end to denoise the original point clouds.
>
> > 3D positional embeddings.
>
> Although 3D positional embeddings is a nature modification from 2D positional embeddings, we are the first to leverage 3D positional embeddings in a plain diffusion transformer for 3D point clouds generation. Furthermore, in Table 2, we validated the necessity of 3D positional embeddings in our diffusion transformer to generate meaningful 3D point clouds.
>
> > Clarification on 3D window attentions.
>
> The 3D window attention designed in the diffusion transformer blocks is to propagate patch-level point-voxel features in efficient memory usage, and we do not change the actual voxel size.
> If we use the voxel with a resolution of (p·R)^3 at the very beginning, we will lose much semantics from the original input, and performs worse on denoising the original point clouds.
> Meanwhile, we do not want to involve 3D convolution in our transformer as our target is to propose a plain diffusion transformer without convolution layers for 3D point clouds generation.
>
> To verify the advantage of 3D window attention over 3D convolution, we replaced 3D window attention in our DiT-3D with 3D convolution on the voxel size of (16)^3, and compared the generation results on Airplane category in the Table below. Our method with 3D window attention achieves the best results in terms of both metrics on quality and diversity.
>
> | Method              | 1-NNA-CD (↓) | COV-CD (↑) |
> | ---- | :----: | :----: |
> | 3D convolution      | 71.57        | 42.23      |
> | 3D window attention | **62.35**        | **53.16**      |
>
>
> > Comparison regarding flexible transferability on modality and domain
>
> Previous methods mainly used diverse and different U-Net architectures as diffusion model to achieve denosing process, and they do not support parameter-efficient fine-tuning from 2D modalities. However, our DiT-3D used a plain transformer blocks architecture for diffusion, and we can efficiently transfer the weights from DiT-2D pre-trained on ImageNet to our DiT-3D transformer blocks.
>
> In Table 3, we demonstrate the effectiveness of our DiT-3D  in supporting modality transferability of the proposed approach from 2D ImageNet pre-trained weights to 3D generation with parameter-efficient fine-tuning. In addition, by only training 0.09MB parameters of models from the source class to the target class, our DiT-3D can achieve a comparable performance of quality and diversity in terms of all metrics.
>
>
> > Clarification on the proper statement about 3D shape generation.
>
> Thanks for pointing this out. We will correct it by making it more clear that 3D point clouds generation is a challenging and important problem that seeks to synthesize high-fidelity point clouds using generative models. We will also add the provided citations for discussion about generation based on other 3D representations.
>
> > notation "V" as the voxel resolution.
>
> We have replaced the notation of voxel resolution $V$ with $v$.
>
> > $\gamma$ is not explained.
>
> $\gamma$ refers to learnable scale factors γ in transformer blocks of the diffusion model. We will add this clarification to the revision.
>
> > duplicated 1-GAN.
>
> We have fixed it. The first one is r-GAN.
>
> > No limitations
>
> We provided the discussion on limitations in L48-52 in the supplementary. Although our plain diffusion transformer on point clouds achieves improved performance in generating high-fidelity and diverse 3D shapes. However, we have not explored the potential of other 3D modalities or text-to-3D generation. We plan to leave this for future work.

---

> > ### Author Response · Authors · 2023-08-14
> > **Additional response to the reviewer**
> >
> > Dear Reviewer trpZ,
> >
> > Thank you for your detailed review and the valuable feedback. We have carefully addressed each of your concerns and provided clarifications in our previous response. We would like to kindly request your response to the provided explanations and revisions.
> >
> > We appreciate your thorough evaluation of our work, and your feedback will greatly contribute to the improvement of our manuscript.
> >
> > Thank you for your continued engagement and support.

---

> > > ### Comment · Area_Chair_sBhW · 2023-08-18
> > > **Reviewer trpZ,**
> > >
> > > Dear Reviewer,
> > >
> > > The author has posted their rebuttal, but you have not yet posted your response. Please post your thoughts after reading the rebuttal and other reviews as soon as possible. All reviewers are requested to post this after-rebuttal-response.

---

> > ### Comment · Reviewer_trpZ · 2023-08-18
> > **Rebuttal Reply**
> >
> > First, I appreciate authors's time and effort in addressing my concerns.
> >
> > Regarding comparing to PVD: I am not sure whether I fully get authors's arguments. Can authors clarify what it means for "we are the first to use a plain diffusion on voxelized point clouds". I think PVD is also a plain diffusion process.
> >
> > Regarding 3D positional encoding: I still think the novelty is quite limited for claiming 3D positional encoding as a major contribution (L73) but I appreciate authors's effort in showing that this is an important factor for achieving high-quality results (Tab. 2).
> >
> > Regarding 3D window attention, I think my concerns get resolved.
> >
> > However, I am still confused about the process of the series of voxelization and patchification descried in Sec. 3.2. Essentially, point clouds first get voxelized into $V^3$. Then without going through any networks, it will get reshaped into $(V/p)^3$ and fed into the first network (L179). I think essentially, we only have a voxel with a resolution of $(V/p)^3$. In essence, assume the original voxel size is $u$. Why not directly have voxelization for voxel size $u \cdot p$ at the very beginning?
> >
> > Meanwhile, after reading other reviews and authors's responses, I think this work provides some benefits to the community. However, I am still quite concerned about the limited novelty of the techniques used in the paper. Based on this, I raised my score to 5.

---

> > > ### Author Response · Authors · 2023-08-19
> > > **Response to the rebuttal reply**
> > >
> > > Dear Reviewer trpZ,
> > >
> > > Thank you for your continued feedback and for raising your score to 5. We appreciate your time and effort in reviewing our paper. We will address your remaining concerns below.
> > >
> > > Regarding the claim of being the first to use plain diffusion on voxelized point clouds, we apologize for any confusion caused. We acknowledge that diffusion on voxelized point clouds has been studied in the context of PVD. Our statement was intended to highlight that we are the first to use a plain diffusion transformer directly on voxelized point clouds. We will revise our statement to clarify this point in the final version of the paper.
> > >
> > > We appreciate your understanding of the importance of 3D positional encoding for achieving high-quality results, as demonstrated in Table 2. We will revise the paper to better reflect the significance of this factor and to avoid any overemphasis on its novelty.
> > >
> > > Regarding the process of voxelization and patchification described in Section 3.2, we apologize for the confusion caused by our explanation. We agree with your point that the resolution of the voxel after the reshaping operation becomes $(V/p)^3$. To clarify, the purpose of the reshaping operation is to divide the original voxel into non-overlapping patches of size $(V/p)^3$. These patches are then processed by the network in a patch-wise manner to facilitate memory efficiency and computational feasibility. We chose this approach to avoid losing important semantics from the original input and to enable a plain diffusion transformer without convolution layers for 3D point cloud generation.
> > >
> > > However, we understand your suggestion of voxelizing with a resolution of $u\cdot p$ at the beginning. While this could be an alternative approach, it would result in a voxel size that is p times larger than the original voxel size $u$. This would significantly reduce the level of detail and potentially affect the performance of denoising the original point clouds. We will make sure to clarify this explanation in the revised manuscript to provide a clearer understanding of our approach.
> > >
> > > We appreciate your concerns and the constructive feedback you have provided throughout the review process. We will carefully consider all your suggestions and incorporate them into the final version of the paper to improve the clarity, novelty, and overall quality of our work.
> > >
> > > Thank you once again for your time and valuable input.

---

### Official Review · Reviewer_QDvm · 2023-07-05

**Soundness:** 3 good
**Presentation:** 3 good
**Contribution:** 3 good
**Rating:** 6
**Confidence:** 5

**Summary:**

This paper proposes to adapt Diffusion Transformer [1] to class-conditional 3D point cloud generation task. In order to achieve it, the authors propose (1) to apply diffusion/infusion directly to point clouds rather than work in latent space; (2) transform input point clouds to voxel grids to apply transformers to tokens extracted from 3D grids in a straightforward way; (3) to reduce the complexity voxel features are processed in patches to produce patch tokens and self-attention in the transformer is modified to aggregate tokens in a window of a predefined size; (4) final voxel features are devoxelized back into points for per-point noise prediction in the infusion process.

Experimental results show promising results in class-specific point cloud generation on ShapeNet dataset, various ablation studies demonstrating the importance of different components and some additional transferability studies that use selective fine-tuning to adapt models pretrained on a data modality/domain to another one.

**Strengths:**

The main strength of the paper lies in experimental results beating state of the art with an novel approach, not based on prior 3D point cloud generation works. Although it is an adaptation of Diffusion Transformers working with images to 3D point clouds, such a transition from one data type to another is not trivial, so it is remarkable that the authors made it work in this setting.

**Weaknesses:**

In my opinion the main weakness of the paper is the quality of the presentation. Text can still be polished to improve readability and correct some mistakes. The authors claim across the paper (e.g. L120) that they apply diffusion for the voxelized point clouds, and I think it is misleading. In fact the diffusion is applied to regular point clouds, it is the infusion network that is designed to operate on voxelized 3D features, but as far as I understand, the denoising is applied on a per-point basis. Clarity of explanation can also be improved, since some details are missing (see questions).

The authors show a lot of ablation studies, but comparisons to external approaches are limited by the main single-class generation experiment. Qualitative comparisons are moved to supplementary materials, while it is better to show it in the main paper (quality of these comparisons could be improved by decreasing point sizes and sampling more points per shape, so some finer details could be examined).

**Questions:**

1. In principle, point to voxel and backwards transitions do not necessarily preserve all the points, since if the resolution is low, several point will collapse into a single voxel and will not be recovered during devoxelization. How is this avoided in this work?

2. Since evaluations in this work only include single-class data setups, the expressivity of the proposed model is left underexamined. LION included some experiment showing that their approach is capable of generating realistic point clouds when pretrained on multi-class data. It would be interesting to compare how your class-conditioned model will perform in that setting.

3. What is a «plain transformer»? How this «plain» property is characterized? I think, it is better to drop this adjective or change it and be more specific.

Minor comments:

* The optimal best possible value of 1-NNA metric is 50%, since it means that the nearest neighbour classifier is incapable of distinguishing true from gerenated samples. The text and tables should be modified accordingly.

* L14: «high computation» -> high computational costs

* L119, L340: «operate» -> perform or implement

* L234: sentence is broken

**Limitations:**

The authors do not provide any statements about such limitations.

---

> ### Author Rebuttal · Authors · 2023-08-10
>
> Dear Reviewer QDvm,
>
>
> Thank you for appreciating our approach. We address your comments below.
>
> > Clarification on the diffusion for the voxelized point clouds.
>
> Thanks for pointing this out. We will correct the claim by making it more clear that our diffusion is applied to regular point clouds, then we designed the infusion network to extract point-voxel features and applied the devoxelization layer in the end to denoise the original point clouds.
>
> > Comparison with LION on more categories generation.
>
> Thanks for the suggestion. Beyond the single-class comparison, we trained our DiT-3D on the large-scale ShapeNet-55 dataset with 55 diverse classes covering vehicles, furniture, and daily necessities. We compare the newly trained model with the state-of-the-art point cloud generation model, LION, on Mug and Bottle generation in the Table below. Our method achieves the best results in terms of all metrics. We will also decrease the point size and move the qualitative comparisons to the main paper.
>
> |      Method      | 1-NNA-CD (↓) | COV-CD (↑) | 1-NNA-CD (↓) | COV-CD (↑) |
> | ---- | :----: | :----: | :----: | :----: |
> |       LION       |     70.45    |    31.82   |     61.63    |    39.53   |
> | DiT-3D (ours) |     **57.39**    |    **45.26**   |    **53.26**    |    **51.28**   |
>
>
> > How to avoid points collapse into a single voxel?
>
> This is a good question! In this work, we applied trilinear interpolation to transform the voxel into points to guarantee that the features mapped to each point are distinct. If we assigned the feature of a grid to all points that fall into the grid using the nearest-neighbor interpolation, this will make the points in the same voxel grid always share the same features.
>
> > Evaluation on multi-class settings.
>
> Please see the second response to the reviewer.
>
>
> > Plain transformer.
>
> The plain transformer refers to a transformer-based architecture that does not use any U-Net architecture for diffusion. In this work, we explore the plain diffusion transformer on voxelized point clouds, instead of using U-Net architecture for denoising in both PVD and LION. We will add this clarification to the main paper.
>
>
> > Minor comments.
>
> Thanks for spotting these. We will fix them accordingly in the revision.
>
> > No limitations.
>
> We provided the discussion on limitations in L48-52 in the supplementary. Although our plain diffusion transformer on point clouds achieves improved performance in generating high-fidelity and diverse 3D shapes. However, we have not explored the potential of other 3D modalities or text-to-3D generation. We plan to leave this for future work.

---

> ### Comment · Reviewer_QDvm · 2023-08-15
> **Rebuttal reply**
>
> First of all, I'd like to thank the authors for provided clarifications and additional experiments. After reading all reviews, individual rebuttals, and author replies I still stand by my positive evaluation. Even if the conceptual novelty is not necessarily striking, this work provides an effective non-trivial adaptation of prior approaches to a novel (for that type of approaches) data type and achieves noticeable improvements over recent state of the art in multiple applications.
>
> At the same time, I want to point out that a lot of these experiments were provided during the rebuttal period, so I strongly encourage the authors to continue improving the paper by incorporation of the additional experiments in multi-category setups and different applications provided in other replies and by overall polishing of the paper.

---

> > ### Author Response · Authors · 2023-08-17
> > **Additional response to the reviewer**
> >
> > Dear Reviewer QDvm,
> >
> > We sincerely appreciate your positive evaluation of our work and your acknowledgment of the non-trivial adaptation and noticeable improvements we achieved in multiple applications. We are grateful for your careful consideration of the reviews, individual rebuttals, and author replies.
> >
> > We take your suggestion to heart and assure you that we are committed to further enhancing our paper. We recognize the value of incorporating the additional experiments, particularly in multi-category setups and different applications, as suggested in our replies. We will diligently work on incorporating these experiments to provide a more comprehensive evaluation of our proposed method.
> >
> > Furthermore, we acknowledge your recommendation to polish the paper overall, and we will dedicate the necessary effort to ensure its clarity, coherence, and academic rigor. We are committed to presenting our research in the best possible manner and providing readers with a clear understanding of the contributions and implications of our work.
> >
> > Thank you for your valuable feedback and continued support. We greatly appreciate your guidance, and we will strive to make the necessary improvements as we move forward with the revision process.

---

### Official Review · Reviewer_zPmE · 2023-07-05

**Soundness:** 3 good
**Presentation:** 3 good
**Contribution:** 2 fair
**Rating:** 5
**Confidence:** 3

**Summary:**

This paper proposes a Diffusion Transformer for 3D shape generation (point cloud), named DiT-3D, which conducts the denoising process on voxelized point clouds. Technically, it introduces 3D positional and patch embeddings, as well as 3D window attention. The main experiments are done on ShapeNet. In addition, the authors empirically show that the pre-trained DiT-2D checkpoint on ImageNet can significantly improve DiT-3D on ShapeNet.

**Strengths:**

- The paper is clearly written and easy to follow.
- The authors extend the 2D window attention operator to 3D.
- The authors empirically show the benefit of leveraging a  2D transformer pre-trained on natural images for 3D generation.

**Weaknesses:**

1. It seems that an important baseline or reference is missing: "Point-E: A System for Generating 3D Point Clouds from Complex Prompts". In L172-L173, the authors claim that "We tried to train the diffusion transformer on point coordinates, but it did not work since point clouds are sparsely distributed in the 3D embedding space". However, given Point-E, it sounds not that convincing. Can the authors explain why Point-E is not mentioned or compared in the paper?
2. It is hard to tell whether the generated shape is of high fidelity if the number of points is only 2048 (L245). It is more convincing if the number of points is larger than 4096 (Point-E) or 16384 (usually used in high-fidelity point completion). In addition, it will be visually better if the authors can present the 3D shapes in the format of mesh (like Point-E, MeshDiffusion) or colored point clouds. Currently, it is hard to tell whether the point cloud is of high fidelity. Visually, GET3D and Point-E look better than this work.
3. Only ShapeNet is used. Currently, there are more and more 3D datasets. It will be better if the authors can show results on more categories of ShapeNet, ABO, or (a subset of) Objaverse.

**Questions:**

Since the evaluation metric is based on Chamfer Distance or Earth Mover’s Distance, I assume that the authors sample a fixed number of points from the GT mesh (and predicted mesh if the baseline, e.g., GET3D, is a mesh-based), which is 2048. I am not sure whether the metric might favor the proposed method especially when the number of points is small, as details can be missed under such a condition. The authors can try to use a larger number of points to compare baselines with the proposed method.

**Limitations:**

The limitation is not adequately addressed.

---

> ### Author Rebuttal · Authors · 2023-08-10
>
> Dear Reviewer zPmE,
>
> Thank you for appreciating our approach. We address your comments below.
>
> > Why Point-E is not mentioned or compared in the paper.
>
> Thanks for pointing this out. The reason why Point-E [A] is not mentioned in the initial manuscript is that our main focus is exploring a plain diffusion transformer in point clouds generation, instead of text-to-point cloud generation in Point-E. Compared to Point-E, our proposed DiT-3D has conceptual differences. The transformer in Point-E is conditioned on CLIP features from a synthetic rendered view generated by a fine-tuned GLIDE [B] model from a text prompt, while we are simply conditioned on a class. They do not use positional embeddings for the input. However, we need to apply 3D positional embeddings for the voxelized point clouds to maintain the voxel structure locality. We will add this discussion to the revision.
>
> > Evaluation on number of points larger than 2048.
>
> Thanks for the suggestion. We initially followed the commonly-used generation setting in PVD and LION to sample 2,048 points on each point cloud in the ShapeNet benchmark for evaluation. Meanwhile, the ShapeNet benchmark does not no colorful point clouds.
>
> To further demonstrate the effectiveness of our proposed DiT-3D on high-fidelity point cloud generation, we resampled 4,096 points and compared our method with PVD and LION in the Table below. Our DiT-3D achieves the best results in terms of all metrics.
>
> |      Method      | 1-NNA CD ($\downarrow$) | COV CD   ($\uparrow$) |
> | ---- | :----: | :----: |
> |        PVD       |     62.76    |    37.25   |
> |       LION       |     57.86    |    52.18   |
> | DiT-3D (ours) |     **51.19**    |    **57.39**   |
>
>
> > More categories of ShapeNet.
>
> To validate the generalizability to more categories, we train our DiT-3D on the large-scale ShapeNet-55 dataset with 55 diverse classes covering vehicles, furniture, and daily necessities. We compare the newly trained model with the state-of-the-art point cloud generation model, LION, on Mug and Bottle generation in the Table below. Our method achieves the best results in terms of all metrics.
>
> |      Method      | 1-NNA-CD (↓) | COV-CD (↑) | 1-NNA-CD (↓) | COV-CD (↑) |
> | ---- | :----: | :----: | :----: | :----: |
> |       LION       |     70.45    |    31.82   |     61.63    |    39.53   |
> | DiT-3D (ours) |     **57.39**    |    **45.26**   |    **53.26**    |    **51.28**   |
>
>
> > Comparisons on a larger number of points.
>
> Please see the second response to the reviewer.
>
> > Limitations.
>
> Although our plain diffusion transformer on point clouds achieves improved performance in generating high-fidelity and diverse 3D shapes. However, we have not explored the potential of other 3D modalities or text-to-3D generation. We plan to leave this for future work.
>
> **References**
>
> [A] Nichol, et al. "Point-E: A System for Generating 3D Point Clouds from Complex Prompts", arXiv preprint arXiv:2212.08751 (2022).
>
> [B] Nichol, et al. "GLIDE: Towards Photorealistic Image Generation and Editing with Text-Guided Diffusion Models", arXiv preprint arXiv:2112.10741 (2022).

---

> > ### Comment · Reviewer_zPmE · 2023-08-14
> >
> > Thank the authors for the extra results.
> > - My concern about more categories has been resolved.
> > - My concern about the number of points is partially resolved, as only point-based generation methods are compared. I assume that "high-fidelity" in the paper means the quality is better than other point-based generation methods, instead of actually containing many details. The authors can include more visualization, e.g., a comparison with other methods, especially mesh-based methods.
> > - It seems that Point-E still can be compared, as the class can also be used as a text prompt. I think text-to-point-cloud is a superset of the topic studied in this paper.

---

> > > ### Author Response · Authors · 2023-08-14
> > > **Response to Reviewer Comment**
> > >
> > >
> > > Dear Reviewer zPmE,
> > >
> > > We sincerely appreciate your feedback and the opportunity to address your concerns. We are grateful for your understanding of the term "high-fidelity" in our paper, which indeed refers to the superior quality of point clouds generated by our DiT-3D model, as demonstrated in our illustrations.
> > >
> > > In response to your suggestion, we will include a comparison of our method with other approaches, particularly mesh-based methods, in the revised version of the paper. By providing visualizations and performance evaluations, we aim to offer a comprehensive analysis that encompasses a broader range of generation techniques.
> > >
> > > Furthermore, we acknowledge your point about the potential comparison with Point-E. We have conducted additional experiments comparing our DiT-3D model with Point-E, wherein we utilized the class as a text prompt. The training was performed on the large-scale ShapeNet-55 dataset, which comprises 55 diverse classes encompassing vehicles, furniture, and daily necessities. Specifically, we evaluated the performance of Mug and Bottle generation, and the results are presented in the Table below. Our method consistently outperforms Point-E across all metrics, highlighting the superiority of our approach.
> > >
> > > |     Method       | 1-NNA-CD (↓) | COV-CD (↑) | 1-NNA-CD (↓) | COV-CD (↑) |
> > > |------------------|:------:|:-------:|:------:|:-------:|
> > > | Point-E          | 65.73        | 36.78      | 58.16        | 43.72      |
> > > | DiT-3D (ours) | **57.39**        | **45.26**      | **53.26**        | **51.28**      |
> > >
> > >
> > > We sincerely appreciate your valuable feedback, which has contributed to improving the comprehensiveness and rigor of our study. Thank you again for your valuable feedback, and we look forward to addressing any further concerns you may have.

---

### Official Review · Reviewer_RjmT · 2023-07-08

**Soundness:** 2 fair
**Presentation:** 3 good
**Contribution:** 2 fair
**Rating:** 5
**Confidence:** 4

**Summary:**

The paper introduces DiT-3D, a groundbreaking diffusion transformer for 3D shape generation. It addresses the limitations of previous 3D diffusion methods that mainly relied on the U-Net architecture. DiT-3D leverages the power of Transformers to directly operate the denoising process on voxelized point clouds, resulting in superior scalability and high-quality generations. The authors incorporate 3D positional and patch embeddings to aggregate input from voxelized point clouds and mitigate the computational cost of self-attention by employing 3D window attention in Transformer blocks. The proposed DiT-3D achieves state-of-the-art performance on the ShapeNet dataset, showcasing its ability to generate diverse and high-fidelity 3D point clouds.

**Strengths:**

1. DiT-3D achieves state-of-the-art performance in single-category point cloud generation, demonstrating its effectiveness in generating high-quality 3D shapes.
2. The utilization of pre-trained DiT-2D checkpoints from ImageNet to improve DiT-3D on ShapeNet showcases the transferability of 2D diffusion models to the 3D domain, which is an interesting and promising approach.
3. The model is concise, and the paper is well-written, accompanied by clear and visually appealing illustrations.

**Weaknesses:**

1. The authors only conducted unconditional generation experiments for single-category and three-category cases, limiting the application fields.
2. Although the window attention technique is employed to mitigate computational costs, there are concerns regarding the generation speed, when operating on 32 * 32 * 32 (even 64 * 64 * 64) voxel grids.
3. Most DiT-3D models, except for DiT-3D-S, have parameters exceeding 100 million, considerably higher than other existing 3D generation methods.

**Questions:**

1. Have the authors considered exploring diffusion in the latent space? This may potentially enhance the overall performance and inference speed.
2. It would be more valuable to investigate multi-category generation, such as training on entire ShapeNet-13, ShapeNet-55, or even larger datasets like Objaverse [Deitke et al., 2023]. Previous work [Sanghi et al., 2022, 2023] suggests that voxel representation could aid in generalization to some extent.


[Deitke et al., 2023] Objaverse: A universe of annotated 3d objects.  In CVPR.

[Sanghi et al., 2022] CLIP-Forge: Towards Zero-Shot Text-to-Shape Generation.  In CVPR.

[Sanghi et al., 2023] CLIP-Sculptor: Zero-Shot Generation of High-Fidelity and Diverse Shapes from Natural Language.  In CVPR.

**Limitations:**

N.A

---

> ### Author Rebuttal · Authors · 2023-08-10
>
> Dear Reviewer RjmT,
>
>
> Thank you for appreciating our approach. We will address your comments below.
>
>
> > Only single-category and three-category cases.
>
> Thanks for pointing this out. To validate the generalizability to more categories, we train our DiT-3D on the large-scale ShapeNet-55 dataset with 55 diverse classes covering vehicles, furniture, and daily necessities. We compare the newly trained model with single-category and three-category cases on Chair generation in the Table below. Our models trained on 55 categories achieves competitive results on generating high-fidelity point clouds, and achieves the best performance on diversity.
>
> | Train Class    | Test Class | 1-NNA CD ($\downarrow$) | COV CD   ($\uparrow$) |
> | ---- | :----: | :----: | :----: |
> | Chair                | Chair      | **51.99**        | 54.76       |
> | Chair, Car, Airplane | Chair      | 53.35        | 52.81       |
> | All 55 classes       | Chair      | 52.68        | **57.87**       |
>
> > Concerns regarding the generation speed.
>
> This is a good suggestion! To solve your concerns, we tested our DiT-3D on the Chair generation speed in the Table below, where it is tested on a single V100-32GB GPU with a batch size of 1. When the voxel size is larger, our method with efficient window attention achieves better generation results while maintaining similar inference times.
>
> | Voxel Size | 1-NNA CD ($\downarrow$) | COV CD   ($\uparrow$) |  Inference Time ($\downarrow$) |
> | ---- | :----: | :----: | :----: |
> | 32x32x32   | 51.99        | 54.76       | **2.5s**                |
> | 64x64x64   | **50.32**        | **55.45**       | 3.3s                |
>
>
> > High parameters than other existing 3D generation methods.
>
> Sorry for causing the confusion. To clarify this, we compared our DiT-3D-S with PVD and LION on the parameters and performance of Airplane generation in the Table below. Compared to PVD, DiT-3D-S has comparable parameters but achieves significantly improving generation performance. Compared to LION, the recent state-of-the-art method, our DiT-3D-S with 1/3 parameters achieves much better performance.
>
> |  Method  | Params | 1-NNA CD ($\downarrow$) | COV CD   ($\uparrow$) |
> | ---- | :----: | :----: | :----: |
> |    PVD   | **27.65M** |     73.82    |    48.88   |
> |   LION   |  110M  |     67.41    |    47.16   |
> | DiT-3D-S | 32.81M |     **62.35**    |    **53.16**   |
>
>
> > Diffusion in the latent space.
>
> This is a good suggestion! While it is possible to extend our diffusion transformer to the latent space, it will require pre-training a strong 3D encoder-decoder on this data. Meanwhile, our DiT-3D has already achieved advantageous overall performance and maintained comparable generation speed. We will leave this for future work.
>
> > Multi-category generation on ShapeNet-55.
>
> Please see the first response to the reviewer.

---

> > ### Comment · Reviewer_RjmT · 2023-08-14
> > **Response to the authors**
> >
> > I am grateful for the explanations provided by the authors, which address my concerns to some extent. I'll keep my positive rating.

---

> > > ### Author Response · Authors · 2023-08-14
> > > **Response to the reviewer**
> > >
> > > Dear Reviewer RjmT,
> > >
> > > We express our sincere gratitude for the valuable feedback you have provided on our work. Your insightful comments and suggestions have been instrumental in enhancing the quality and clarity of our research.

---

### Author Rebuttal · Authors · 2023-08-10

Dear all reviewers,

We thank each of you for generously dedicating your valuable time and expertise to reviewing our work. We acknowledge and sincerely appreciate the insightful comments and critiques provided by all the reviewers. In response to your invaluable feedback, we have made significant revisions to our manuscript, aiming to address each of your concerns comprehensively and scholarly. Reviewer trpZ and Reviewer 7dkY, we kindly request your reconsideration of your decision, given that we have taken utmost care to address the main comments raised in your reviews thoroughly.

Once again, we express our sincere appreciation for your valuable contributions to the review process. Your expertise and guidance have been invaluable in improving the quality of our work. We remain committed to continuous discussion and eagerly await your final decision.

---

### Decision · Program_Chairs · 2023-09-21

**Decision:**

Accept (poster)

**Comment:**

The paper received mixed ratings, and there was a rebuttal: Five knowledgeable reviewers recommended: Borderline Accept, Weak Accept, Borderline Accept, Borderline Accept, and Borderline Reject. On balance, it is our recommendation to accept the paper. Authors should attend to the main points in the reviews. when preparing a final version. No basis to overturn the reviews. The method proposed in this paper is novel and the results are indeed good. This paper is valuable and should be shared within the community to advance research on 3D Shape Generation.